# Structured Learning of Compositional Sequential Interventions

Jialin Yu[1]    Andreas Koukorinis[1]    Nicolò Colombo[2]    Yuchen Zhu[1]    Ricardo Silva[1]

[1]University College London    [2]Royal Holloway, University of London

## Abstract

We consider sequential treatment regimes where each unit is exposed to combinations of interventions over time. When interventions are described by qualitative labels, such as "close schools for a month due to a pandemic" or "promote this podcast to this user during this week", it is unclear which appropriate structural assumptions allow us to generalize behavioral predictions to previously unseen combinations of interventions. Standard black-box approaches mapping sequences of categorical variables to outputs are applicable, but they rely on poorly understood assumptions on how reliable generalization can be obtained, and may underperform under sparse sequences, temporal variability, and large action spaces. To approach that, we pose an explicit model for *composition*, that is, how the effect of sequential interventions can be isolated into modules, clarifying which data conditions allow for the identification of their combined effect at different units and time steps. We show the identification properties of our compositional model, inspired by advances in causal matrix factorization methods. Our focus is on predictive models for novel compositions of interventions instead of matrix completion tasks and causal effect estimation. We compare our approach to flexible but generic black-box models to illustrate how structure aids prediction in sparse data conditions.

## 1   Contribution

Many causal inference questions involve a treatment sequence that varies over time. In the (discrete) time-varying scenario, an *unit* $n$ is exposed to a sequence $D_n^1, D_n^2, D_n^3, \ldots$ of *actions*, or *treatments*, potentially controllable by an external agent [22, Ch. 19] intervening on them. Each $D_n^t$ can be interpreted as a cause of the subsequent behavior $X_n^t, X_n^{t+1}, \ldots$, and its modeling amenable to the tools of causal inference [e.g., 35, 36, 31, 17, 11, 49]. For instance, each $D_n^t$ can take values in the space of particular drug dosages which can be given to an in-patient, or combinations of items to be promoted to a user of a recommender system. Here, patients and users are the units, and $X_n^t$ denotes a measure of their state or behavior under time $t$. Dependencies can result in a dense causal graph, as in Fig. 1. Further background on this classical problem is provided in Appendix A.

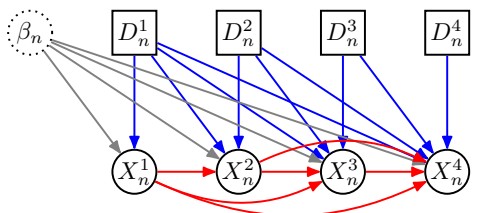

Figure 1: Within unit $n$, actions $D_n^t$ interact with (latent) random effect parameters $\beta_n$ to produce behavior $X_n^t$ represented as a dense graphical model with square vertices denoting interventions $\mathrm{do}(d_n^{1:t})$ [33, 16]. Further assumptions will be required for the identifiability of the impact of interventions and their combination, including how temporal impact takes shape and the number of independent units of observation.

**Scope.** We introduce *an approach to predict the effects of actions within a combinatorial and sparse space of categorical intervention sequences.* That is, for an unit $n$ and given past behavioral sequence $X_n^{1:t-1} := [X_n^1, \ldots, X_n^{t-1}]$, manipulable variables $D_n^{1:t-1} := [D_n^1, \ldots, D_n^{t-1}]$ and time-fixed covariates $Z_n$, we predict future behavior $X_n^t, X_n^{t+1}, \ldots, X_n^{t+\Delta}$ under the (partially hypothetical) interventional regime $\mathrm{do}(d_n^{1:t+\Delta})$. The latter denotes controlling the action variables by external intervention [33]. For mean-squared error losses, prediction means learning $\mathbb{E}[X_n^{t+\delta} \mid x_n^{1:t-1}, z_n, \mathrm{do}(d_n^{1:t+\delta})]$, $0 \leq \delta \leq \Delta$. We consider the case where sequential ignorability holds, by randomization or adjustment [33, 22], in order to focus on predictive guarantees for previously unseen action sequences.

**Challenge.** Much of the sequential intervention literature in AI and statistics considers generic black-box or parametric models[1] to map sequences $D_n^{1:t} := [D_n^1, D_n^2, \ldots, D_n^t]$ to behavior $X_n^t$ and beyond, sometimes exploiting strong Markovian assumptions or short sequences and small action spaces [11, 43]. An unstructured black-box for sequences, such as recurrent neural networks and their variants [e.g. 23, 13, 45], can learn to map a sequence to a prediction. This is less suitable when actions are categorical and *sparse*, in the sense that most entries correspond to some baseline category of "no changes". That is, there is no obvious choice of smoothing or interpolation criteria to generalize from seen to unseen strings $D_n^{1:T}$, with the practical assumption being that we observe enough variability to reliably perform this interpolation. There is an increased awareness of problems posed by large discrete spaces among pre-treatment covariates [47, 48], which is of a different nature as losses can be minimized with respect to an existing natural distribution, as opposed to the interventional problem where the action variables are allowed to be chosen and so sparsity conditions can easily result in very atypical test cases of interest. Several papers address large intervention spaces (mostly for non-sequential problems) without particular concerns about non-smooth, combinatorial identifiability [25, 32, 38, 39]. A few more recent papers address explicit assumptions of identifiability directly in combinatorial categorical spaces by sparse regression [2] or energy functions [8], but without a longitudinal component. More classically, tools like the do-calculus [33] and their extensions [15] can be used with directed acyclic graphs (DAGs) to infer combinatorial effects, although they often restrict each manipulated variable to directly affect only a small set of variables (e.g. [5]). With strong assumptions and computational cost, this may include latent variables [50]. Alternative representation learning ideas for carefully extrapolating to unseen interventions are presented by [37] in the non-sequential, non-combinatorial, setting.

In this paper, we consider the conservative problem of *identification guarantees* for the effect of sequential combinations of interventions which may never have co-occurred jointly in training data, *formalizing assumptions* that allow for the transfer and recombination of information learned across units. These are not of the same nature as identifiability of causal effects from observational data [33] or causal discovery [41], but of *causal extrapolation*: even in the randomized (or unconfounded) regime, it is not a given that the distribution under a particular regime $\mathrm{do}(d_n^{1:t})$ can be reliably inferred from limited past combinations of action sequences without an explicit structure on the causal generative model. We will consider the case where actions imply *non-stationary transitions*: once unit $n$ is exposed to $D_n^t = d_n^t$, then $X_n^{t+\Delta}$ will have transition dynamics that can be affected not only by the choice of $d_n^t$ but also the difference $\Delta$, resulting in dense dependency structures such as the one in Figure 1. As multiple treatments are applied to the same unit $n$, there will be a *composition* of effects that will depend not only on the labels of the actions, but also on the order by which they take place. If a substantive number of choices of treatments at time $t$ is available, a large dataset may have no two sequences $D_n^{1:T}$ and $D_{n'}^{1:T}$ taking the same values, even when considering only the unique treatment values applied to $n$ and $n'$, regardless of ordering and time stamps.

**Problem statement.** Let each individual unit $n$ be described by time-invariant features $Z_n$ and a pair of time-series $(X_n^{1:T}, D_n^{1:T})$. Here, $V^{t_1:t_2}$ denotes a slice of a discrete-time time-series of variables $V^t$ from $t_1$ to $t_2$, inclusive. That observations for all units stop at time $T$ is not a requirement, but just a way of simplifying presentation. Each $D_n^t$ is a categorical action variable which, when under external control, represents a potential *intervention*, encoded by values in $\mathbb{N}$. We assume data is available under a sequentially ignorable selection of actions [33, 22], either by randomization or consequences of an adjustment. $D_n^t$ describes a (possibly hypothetical) intervention which causes changes to the distribution of $X_n^{t:\infty}$. The special value $D_n^t = 0$ denotes an "idle" or "default" treatment level that can be interpreted as "no intervention". We are given a training set of units

---

[1]We include here models which explicitly parameterize causal contrasts in longitudinal interventions, such as the blip models of [36].

$(Z_n, X_n^{1:T}, D_n^{1:T})$. The goal is to predict future sequence of behavior $X_{n^\star}$ for a (training or test) unit $n^\star$ under a hypothetical intervention sequence $\mathrm{do}(d_{n^\star}^{1:T+\Delta})$. This process takes place under the stable unit treatment value assumption (SUTVA) [24], meaning that action variables $D_n^t$ only affect unit $n$.

Each intervention on $D_n^t$ is allowed to have an impact on the whole future series $t, t+1, t+2, \ldots$. It can correspond to an instantaneous shock ("give five dollars of credit at time $t$") or an action that takes place over an extended period of time ("promote this podcast from time $t$ to time $t+5$"): we just assume the meaning to be directly encoded within arbitrary category labels $0, 1, 2, 3, \ldots$. In a real-world scenario where a same action can be applied more than once, different symbols in $\mathbb{N}$ should be used for each instance (e.g., 1 arbitrarily standing for "give five dollars", and 2 for "give five dollars a second time" and so on).

In Section 2, we provide an account of what we mean by compositionality of interventions, with explicit assumptions for identifiability. In Section 3, we describe an algorithm for likelihood-based learning, along with approaches for predictive uncertainty quantification. Further related work is covered in Section 4 before we perform experiments in Section 5, highlighting the shortcomings of black-box alternatives.

## 2 A Structural Approach for Intervention Composition

For simplicity, we will assume scalar behavioral measurements $X_n^t$, as the multivariate case readily follows. Consider the following conditional mean model for the regime $\mathrm{do}(d_n^{1:t})$,

$$\mathbb{E}[X_n^t \mid x_n^{1:t-1}, z_n, \mathrm{do}(d_n^{1:t})] = (\phi_n^t)^\mathsf{T}(\beta_n \odot \psi_n^t) = \sum_{l=1}^r \phi_{nl}^t \beta_{nl} \psi_{nl}^t, \tag{1}$$

where $\odot$ is the elementwise product, with $\phi_n^t$, $\beta_n$ and $\psi_n^t$ defined below. We postpone explaining the motivation for this structure to Section 2.1. First, we describe this structure in more detail.

Using $p(\cdot)$ to denote generic probability mass/density functions for random variables given by context, assume $p(x_n^t \mid \mathrm{do}(d_n^{1:t'})) = p(x_n^t \mid \mathrm{do}(d_n^{1:t}))$ for any $t' \geq t$ (future interventions do not affect the past), and $p(z_n \mid \mathrm{do}(d_n^{1:t})) = p(z_n)$ for all $t$. We do not explicitly condition on $D_n^{1:t}$ in most of our notation, adopting a convention where intervention indices $\mathrm{do}(d_n^{1:t'})$ are always in agreement with the (implicit) corresponding observed $D_n^{1:t} = d_n^{1:t}$. We define $\phi_{nl}^t := \phi_l(x_n^{1:t-1}, z_n)$ as evaluations of basis functions $\phi_l(\cdot)$. For now, we will assume such $\phi_l(\cdot)$ functions to be known. Each unit $n$ has individual-level coefficients $\beta_n$. Causal impact, as attributed to $\mathrm{do}(d_n^{1:t})$, is given by

$$\psi_{nl}^t := \prod_{t'=1}^t \psi_l(d_n^{t'}, t', t), \tag{2}$$

where function $\psi_l(d, t', t)$ captures a trajectory in time for the effect of $D_n^{t'} = d$, for $t' < t$. Compositionality will be described in the sequel in two ways: first, as a way of combining the effects of actions in a type of recursive functional composition (*compositional recursion*); second, by interpreting this function composition at time $t$ in terms of an operator warping a baseline unit-level feature vector $\beta_n$ with an interventional-level embedding vector $\psi(d_n^t, t, t)$ (*compositional warping*).

**Effect families.** We consider two variants for $\psi(\cdot)$. The first is a *time-bounded* variant which, for $d > 0$, defines $\psi_l(d, t', t) := w_{dl,t-t'}$, where $w_{dl,t-t'}$ is a free parameter for $0 \leq t - t' < k_d$, otherwise $w_{dl,t-t'} = w_{dl,k_d-1}$. The intuition is that the effect of intervention level $d$ has no shape constraints but it cannot affect the past and it must settle to a constant after a chosen time window hyperparameter $k_d > 0$. The default level $d = 0$ has no effect and no free parameters, with $\psi_l(0, t', t) = 1$. The second variant is motivated by real-world phenomena where causal impacts diminish their influence over time and result in a new equilibrium [9]. We consider a *time-unbounded* shape with exponential decay, parameterized as

$$\psi_l(d, t', t) := \sigma(w_{1dl})^{t-t'} \times w_{2dl} + w_{3dl}, \tag{3}$$

where $\sigma(\cdot)$ is the sigmoid function, so that $0 < \sigma(w_{1d}) < 1$. As $t$ grows, $\sigma(w_{1d})^{t-t'}$ goes to 0. $w_{3d}$ is the stationary contribution of an intervention at level $d$. In particular, for $d = 0$, we define $w_{10l} = w_{20l} = 0$ and $w_{30l} = 1$, while for $d > 0$ we have that $w_{1dl}, w_{2dl}$ and $w_{3dl}$ are free parameters to be learned. Let us interpret the previous $k_d$ hyperparameter as the *dimensionality of intervention level $d$*. As the model of Eq. (3) is given by vectors $w_{1d}, w_{2d}, w_{3d}$, we here define $k_d = 3$.

## 2.1 Representation Power: Compositional Recursion and and Compositional Warping

Eq. (1) is reminiscent of tensor factorization and its uses in causal modeling [7, 3], where the primary motivation comes from the imputation of missing potential outcomes [24] taking place on a pre-determined period in the past. From a predictive perspective, it can be motivated from known results in functional analysis (such as Proposition 1 of [25]) that allows us to represent a function $f(x)$ by first partitioning its input into $(x_a, x_b)$ and controlling the approximation error given by an inner product of vector-valued functions,

$$f(x_a, x_b) \approx f_a^\mathsf{T}(x_a) f_b(x_b), \tag{4}$$

which we call *the factorization trick*. In practice, we choose the dimensionality of such vectors $\{f_a(\cdot), f_b(\cdot)\}$ by a data-driven approach. Although those results apply typically to smooth functions with continuous inputs (such as the Taylor series approximation, which relies on vectors of monomials with coefficients constructed from derivatives), discrete inputs such as $D_n^{1:T}$, at a fixed dimensionality $T$, can be separated from the other inputs as

$$f(n, d^{1:T}, x^{1:T}, z) := f_d^\mathsf{T}(d^{1:T}) f_{nxz}(n, x^{1:T}, z). \tag{5}$$

Here, similarly to classical ANOVA models, $f_d(\cdot)$ is a one-hot encoding vector for the entire (exponentially sized) space of possible $d_n^{1:T}$ trajectories. That is, $f_d(\cdot)$ is binary with exactly one non-zero entry, corresponding to which of the possible trajectories $d_n^{1:T}$ took place. The codomain of $f_{nxz}(\cdot, \cdot, \cdot)$ must be a set of vectors of corresponding (exponential in $T$) length.

In what follows, we will start from the premise of further partitioning $f(n, d^{1:T}, x^{1:T}, z) := \mathbb{E}[X_n^{T+1} \mid x_n^{1:T}, z_n, \mathrm{do}(d_n^{1:T})]$ by applying the factorization trick of Eq. (4) repeatedly, justifying the generality of the right-hand side (RHS) of Eq. (1). We will accomplish this by first introducing more structure into the respective $f_d(d^{1:T})$, based on a recursive composition of functions. We will then show in Proposition 1 how the RHS of Eq. (1) follows. We also provide a complementary theoretical result in Proposition 2 which, by assuming a bounded $T$, describes the choice of dimensionality in the application of Eq. (4) as controlling a formal approximation error for a general class of functions.

**Main results: factorization trick with compositional recursion.** As we do not want to restrict ourselves to fixed $T$ (and $N$), not to mention an exponential cost of representation, we will resort to fixed-dimensionality embeddings of sequences and identities. In particular, we design $f_d(d^{1:t})$ to have a finite and tractable output dimensionality $r_g$ for all $t$. Inspired by sequential hidden representation models such as recurrent neural networks, for each entry $l \in 1, 2, \ldots, r_g$ in the output of $f_d(\cdot)$, we define $f_{dl}(d^{1:t})$ via the following recursive definition:

$$\begin{aligned}
f_{dl}(d^{1:t}) &:= g_l(d^{1:t}, t, t), \\
g_l(d^{1:t'}, t', t) &:= h_l(g_l(d^{1:t'-1}, t'-1, t), d^{t'}, t', t), \quad 0 < t' \leq t, \\
h_l(v, d, t', t) &:= h_{l_1}(v)^\mathsf{T} h_{l_2}(d, t', t).
\end{aligned} \tag{6}$$

The base case for $t' = 0$ is $g_l(d^{1:0}, 0, t) \equiv 1$ for all $t$, with $d^{1:0}$ undefined. Moreover, we define $h_l(v, 0, t', t) \equiv v$ for any $(v, t', t)$ – that is, intervention level $D_n^{t'} = 0$ does nothing.

In order to define $h_l(v, d, t', t)$ for $d > 0$, we will apply the same trick of splitting the input into two subsets, and defining a vector representation on each. The representational choice $h_l(v, d, t', t) := h_{l_1}(v)^\mathsf{T} h_{l_2}(d, t', t)$ requires a choice of dimensionality $r_{h_l}$ for the inner vectors $h_{l_1}, h_{l_2}$. Recursively applying (6) in this product form can still be done relatively efficiently, but choosing $r_{h_l} = 1$ for all $l$ simplifies matters. Finally, we apply the factorization trick one last time to define $f_{nxz}(n, x^{1:T}, z)$ within Eq. (5) to get to a factorization form analogous to Eq. (1). More formally:

**Proposition 1** *Let* **(i)** $f(n, d^{1:T}, x^{1:T}, z) := f_d^\mathsf{T}(d^{1:T}) f_{nxz}(n, x^{1:T}, z)$, *where function sequences* $f_d(d^{1:T})$ *and* $f_{nxz}(n, x^{1:T}, z)$ *are defined for all* $T \in \mathbb{N}^+$ *and have codomain* $\mathbb{R}^{r_g}$; **(ii)** $f_{dl}(d^{1:T})$ *be given as in Eq. (6) with* $r_{h_l} = 1$; **(iii)** *the l-th entry of* $f_{nxz}(n, x^{1:T}, z)$ *be given by* $f_{nxzl}(n, x^{1:T}, z) := u_l(n)^\mathsf{T} v_l(x^{1:T}, z)$, *where both* $u_l(\cdot)$ *and* $v_l(\cdot, \cdot)$ *have codomain* $\mathbb{R}^{r_{s'}}$; **(iv)** $v_l(x^{1:t}, z)$ *be defined analogously to* $f_{dl}(d^{1:t})$ *as in Eq. (6), carrying out the fixed-size z as an extra argument. Then there exists some integer* $r$ *and three functions* $a, b$ *and* $c$ *with codomain* $\mathbb{R}^r$ *so that* $f(n, d^{1:T}, x^{1:T}, z) = \sum_{l=1}^r a_l(x^{1:T}, z) \times b_l(n) \times c_l(d^{1:T}, T)$. *Moreover* $c_l(d^{1:T}, T) = \prod_{t=1}^T m_l(d^t, t, T)$ *for some function* $m_l : \mathbb{N}^3 \to \mathbb{R}$.

Proofs of this and next results are given in Appendix B. The above shows that if we start with an assumption that the ground truth function takes some desired form, then there is some finite $r$ for which our combination of basis functions can parameterize the ground truth function for arbitrary $T$, analogous to Eq. (1), where $(a, b, c)$ plays the role of $(\phi, \beta, \psi)$. On the other hand, we show a companion result below: that if we hold $T$ fixed, there is some fixed value for $r$, which depends polynomially on $T$, such that any measurable ground truth function can be approximated by our combination with some appropriate choice of basis functions.

**Proposition 2** *Given a fixed value of $T$, suppose we have a measurable function $f : \bigcup_{t=2}^{T} \mathcal{X}^{1:t-1} \times \bigcup_{t=1}^{T} \mathcal{D}^t \times \mathcal{Z} \to \mathbb{R}^d$, $t \leq T$, where $\mathcal{X}^{1:t-1}$ and $\mathcal{Z}$ are compact, $\mathcal{D}^t$ is finite. Let $\mathbf{x} \in \bigcup_{t=2}^{T} \mathcal{X}^{1:t-1}$ and let $\mathbf{d} \in \bigcup_{t=1}^{T} \mathcal{D}^t$. For $d = 1, \cdots, d_{max}$, let $m_d : \bigcup_{t=1}^{T} \mathcal{D}^t \to \{1, \cdots, T\}$ with $m_d(\mathbf{d}) = t$ if $\exists t$ s.t. $\mathbf{d}_t = d$ else $0$. Then, for any $\epsilon > 0$, there exist $r = O(T^{d_{max}+1})$ and measurable functions $\phi_l : \mathcal{X}^{1:t-1} \times \mathcal{Z} \to \mathbb{R}^d$ and $\psi_{ld'} : \mathcal{D}^t \to \mathbb{R}$, and real numbers $\beta_l$, such that*

$$sup_{\mathbf{x}\in\bigcup_{t=2}^{T}\mathcal{X}^{1:t-1}, \mathbf{d}\in\bigcup_{t=1}^{T}\mathcal{D}^t, z\in\mathcal{Z}} \left| f(\mathbf{x}, \mathbf{d}, z) - \sum_{l=1}^{r} \phi_l(\mathbf{x}, z) \times \beta_l \times \prod_{d=1}^{d_{max}} \psi_{ld'}(m_{d'}(\mathbf{d})) \right| \leq \epsilon. \quad (7)$$

**Notes.** The theoretical results above are reminiscent of the line of work on matrix factorization (e.g., Appendix A of [4]). Eq. (1) and Eq. (2) are motivated by the success of approaches for representation learning of sequences, which assume that we can compile all the necessary information from the past using a current finite representation at any time step $t$. However, Propositions 1 and 2 further imply that, for $f(n, d^{1:T}, x^{1:T}, z) := \mathbb{E}[X_n^T \mid x_n^{1:T-1}, z_n, \mathrm{do}(d_n^{1:T})]$, the representation of functions in a multilinear combination of adaptive factors (Eq. (1)) is motivated by more fundamental results in functional analysis. Our assumption that no value of $d$ (other than the baseline 0) can be used more than once has mostly an empirical motivation, as this would imply unrealistic assumptions about the same intervention having the same effect when applied multiple times. Ultimately, we assume that the factors $h_{l_2}(d, t', t)$ leading to Eq. (2) are time-translation invariant, being a function of $d$ and $t - t'$ only. While it is possible to generalize beyond translation invariant models, as done in the synthetic controls literature [1] that inspired causal matrix factorization ideas [7, 3, 4], this would complicate matters further by requiring a fourth factor into the summation term of Eq. (1) encoding absolute time. Models for time-series forecasting with parameter drifts can be tapped into and integrated with our model family, but we leave these out as future work.

**Interpretation: compositional warping.** At the start of the process, where all units are assumed to be at their "natural" state (assuming $D_n^1 = 0$), we can interpret $\eta_n^1 := \beta_n$ as a finite basis representation, with respect to $\phi_n^1$, of $\mathbb{E}[X_n^1 \mid Z_n, \mathrm{do}(d_n^1 = 0)] = (\phi_n^1)^{\mathsf{T}}\eta_n^1$. Each $\eta_{nl}^1$ can be seen as a latent feature of unit $n$. Effect vector $\psi_n^t$ defines a *warping* of $\eta_n^{t-1}$ that describes how treatments affect the behaviour of an individual in feature space, with the resulting $\eta_n^t := \eta_n^{t-1} \odot \psi_n^t$. The modifier $\psi_n^t$ can be interpreted as reverting, suppressing or promoting particular latent features that are assumed to encode all information necessary to reconstruct the conditional expectation of $X_n^t$ from the chosen basis (notice that intervention level 0 implies $\eta_n^t = \eta_n^{t-1}$ since we define $\psi_l(0, t', t) = 1$). This presents a sequential warping view of compositionality of interventions. Our framing also conveniently reduces the problem of identifiability of conditional effects to the problem of identifying baseline vectors $\beta_n$ and warp function embeddings $\psi_l(d, t', t)$, as we will see next.

## 2.2 Identification and Data Assumptions

Assume for now that functions $\phi$ are known. As commonly done in the matrix factorization literature, assume also that we have access to some moments of the distribution. In particular, for given $N$ data points and $T$ time points, we have access to $\mathbb{E}[X_n^t \mid x_n^{1:t-1}, z_n, \mathrm{do}(d_n^{1:t})]$. We will impose conditions on the realizations of $X_n^{1:T}$ and values chosen for $d_n^{1:T}$, as well as values of $N$ and $T$ as a function of $r$, in order to identify each $\beta_n$ and the parameters of $\psi_l(d, t', t)$ for all intervention levels $d$ of interest. The theoretical results presented, which describe how parameters are identifiable from *population* expectations and *known function spaces* determined by a prescribed basis $\phi$ of *known and finite dimensionality $r$*, will then provide the foundation for a practical learning algorithm in the sequel.

**Assumption 1** *For a given individual $n$, we assume there is an initial period $T_{0_n} \geq r$ with "no interventions" i.e., $D_n^t = 0$ for all $1 \leq t \leq T_{0_n}$.*

**Assumption 2** *Let $A_n$ be a $(T_{0_n} - 1) \times r$ matrix, each row $t = 1, 2, \ldots, T_{0_n} - 1$ is given by $\phi_n(x_n^{1:t}, z_n)^\mathsf{T}$ realizations under regime $\mathrm{do}(d_n^{1:t})$. Let $A_n$ be full rank with left pseudo-inverse $A_n^+$.*

The first assumption can be interpreted as allowing for a "burn-in" period for unit $n$ under only the default action. The second assumption ensures enough diversity among realized features $\phi_n$ so that a least-squares projection can be invoked to identify $\beta_n$, as formalized in the following proposition.

**Proposition 3** *Let $b_n$ be a $(T_{0_n} - 1) \times 1$ vector with entries $\mathbb{E}[X_n^{t+1} \mid x_n^{1:t}, z_n, \mathrm{do}(d_n^{1:t+1})]$. Under Assumptions 1 and 2, $\beta_n$ is identifiable from $A_n$ and $b_n$.*

The next assumption is the requirement that any action level $d$ of interest is carried out across a large enough number of units. Moreover, they must be sufficiently separated in time from follow-up non-default action levels so that their parameters can be identified. Assumption 4 is a counterpart to Assumption 2, ensuring linear independence of feature trajectories.

**Assumption 3** *For a given intervention level $d$, assume $\psi_l(d, t', t) \neq 0$ for all $l, t, t'$ and that there is at least one time index $t$, and one set $\mathcal{N}_d$ containing all units $n$ where $d_n^{t+1} = d$, such that: (i) $N_d := |\mathcal{N}_d| \geq r$; (ii) $\forall n \in \mathcal{N}_d, d_n^{t+2} = \cdots = d_n^{t+k_d - 1} = 0$, where $k_d$ is the dimensionality of the intervention level $d$.*

**Assumption 4** *Let $n_1, n_2, \ldots, n_{N_d}$ index the elements of a set of units $\mathcal{N}_d$. For $t' = 1, 2, \ldots, k_d - 1$, let $A_{dt'}$ be a $N_d \times r$ matrix where each row $i = 1, \ldots, N_d$ is given by $\phi_{n_i}(x_{n_i}^{1:t+t'-1}, z_{n_i})^\mathsf{T}$ realized under regime $\mathrm{do}(d_{n_i}^{1:t+t'-1})$. We assume that each $A_{dt'}$ is full rank with left pseudo-inverse $A_{dt'}^+$.*

This leads to the main result of this section:

**Theorem 1** *Assume we have a dataset of $N$ units and $T$ time points $(d_n^{1:T}, x_n^{1:T}, z_n)$ generated by a model partially specified by Eq. (1). Assume also knowledge of the conditional expectations $\mathbb{E}[X_n^t \mid x_n^{t-1}, z_n, \mathrm{do}(d_n^{1:t})]$ and basis functions $\phi_l(\cdot, \cdot)$ for all $l = 1, 2, \ldots, r$ and all $1 \leq n \leq N$ and $1 \leq t \leq T$. Then, under Assumptions 1-4 applied to all individuals and all intervention levels $d$ that appear in our dataset, we have that all $\beta_n$ and all $\psi_l(d, \cdot, \cdot)$ are identifiable.*

Notice that nothing above requires prior knowledge of all intervention levels which will exist, and could be applied on a rolling basis as new interventions are invented. There is no need for all units to start synchronously at $t = 1$: the framing of the theory assumes so to simplify presentation, with the only requirement being that each unit is given a "burn-in" period of at least $T_0$ steps and that each new intervention level $d$ is applied to at least $r$ units which have not been perturbed recently by relatively novel interventions. Ultimately, the gap between a novel intervention level and the next one should be sufficiently large according to the complexity of the model, as indexed by $r$. This makes explicit that we do not get free identification: the more complex the domain requirements, the more information we need, both in terms of the time waited and the number of observations.

## 3 Algorithm and Statistical Inference

This section introduces a learning algorithm, as well as ways of quantifying uncertainty in prediction. We also allow for the learning of adaptive basis functions $\phi$. As Eq. (1) does not define a generative model, which will be necessary for multiple-steps-ahead prediction, for the remainder of this section we will assume the likelihood implied by $X_n^t \mid x_n^{1:t-1}, z_n, \mathrm{do}(d_n^{1:t}) \sim \mathcal{N}(f(x_n^{1:t-1}, z_n, d_n^{1:t}), \sigma^2)$. Here, the conditional mean $f(\cdot, \cdot, \cdot)$ is given by Eq. (1). As $\sigma^2$ is not affected by $D$, it can be easily shown to not require further identification results. In general, if parameters in our likelihood are implied by a finite set of estimating equations, we can parameterize it in the multilinear form analogous to (1) and repeat the analysis of the previous section. In practice, the functionals (such as the conditional expectations of $X_t$) used in an estimating equation are unknown. Matrix factorization methods can be used directly by first applying a smoothing method to the data to get plug-in estimates of these functionals [42], but we will adopt instead a likelihood-based approach.

### 3.1 Algorithm: CSI-VAE

We treat each $\beta_n$ as a random effects vector, giving each entry $\beta_{nl}$ an independent zero-mean Gaussian prior with variance $\sigma_\beta^2$. Along with all hyperparameters, we optimize the (marginal) log-

likelihood by gradient-based optimization. With a Gaussian likelihood, the posterior and filtering distribution of each $\beta_n$ can be computed in closed form, but in our implementation we used a black-box amortized variational inference framework [27, 30, 34] anyway that can be readily put together without specialized formulas and is easily adaptable to other likelihoods. We use a mean-field Gaussian approximation with posterior mean and variances produced by a gated recurrent unit (GRU) model [14] composed with a multilayer perceptron (MLP). Therefore, the approximate posterior at $t$ time steps follows from $\mu_{q,\beta,n} := \text{MLP}(\text{GRU}_{\eta_{\beta,1}}(d_n^{1:t}, x_n^{1:t}, z_n))$, and $\log \sigma_{q,\beta,n} := \text{MLP}(\text{GRU}_{\eta_{\beta,2}}(d_n^{1:t}, x_n^{1:t}, z_n))$. Prediction for $X^{t+1:t+\Delta}$ is done by sampling $M$ trajectories, where for each trajectory we first sample a new $\beta_n$ from the mean-field Gaussian approximation. We set each forward $\hat{X}_n^{t+i}$ to be the corresponding marginal Monte Carlo average. Each basis vector $\phi_n^t$ is parameterized via another GRU model such that $\phi_n^t := \text{MLP}(\text{GRU}_\phi(x_n^{1:t-1}, z_n))$. Although the theory in the previous section relies on known basis functions, we could train this GRU up to a threshold time point of $T_0$ and freeze it from that point, each $\beta_n$ being defined as the unit-level coefficient vector under this basis. In practice, we found that doing end-to-end learning provides a modest improvement, and this will be the preferred approach in the experiments. We do find that it is sometimes more stable to condition only up to time $T_0$ for the computation of the variational posterior of $\beta$. Hence, we adopt this pipeline for any results reported in this paper, unless otherwise specified. We call our method *Compositional Sequential Intervention Variational Autoencoder (CSI-VAE)*.

## 3.2 Distribution-free Uncertainty Quantification

Conformal prediction (CP) [46, 19] provides prediction intervals with coverage guarantees. The intervals are computed using a calibration set of labeled samples and include the future samples with non-asymptotic lower-bounded probability. We consider Split Conformal Prediction in two setups.

1. **Hold-out predictions.** We have a set of *historical users*, $n = 1, \ldots, N$, whose behavior has been observed until time $t + \Delta$. The task is to predict the behaviour at time $t + \Delta$ of a *new user*, $n = N + 1$, who has been observed up to time $t$, i.e. to predict $X_{N+1}^{t+\Delta}$ given $X_{N+1}^{1:t}$ and $D_{N+1}^{1:t}$. If we assume we have used the history of the new user $X_n^{1:t}$, $n = 1, \ldots, N + 1$, to train the model, calibration and test samples are exchangeable.

2. **Next-intervention predictions.** We have observed the behavior of *all users*, $n = 1, \ldots, N + 1$, up to time $t$ and aim to predict the effects of the next intervention, which happens at time $t + 1$ for all users, i.e. $D_n^{t+1} \neq 0$, holding $D_n^{t+2:t+\Delta} = 0$. The task is to predict $X_n^{t+\Delta}$ under $\text{do}(d_n^{1:t+\Delta})$ for $n = 1, \ldots, N + 1$, but in what follows we will drop the explicit $\text{do}(\cdot)$ indexing to keep notation lighter, referring explicitly to past observed $D^{1:t}$ only (as its sampling distribution matters), and assuming sequential ignorability by assumption or randomization. Calibration and test are *not exchangeable*, because i) the joint distribution after time $t$, $P_T \sim (X_n^{t+\Delta}, X_n^{1:t}, D_n^{1:t})$ may be different from the one before time $t$, $P_C \sim (X_n^{t'+\Delta}, X_n^{1:t'}, D_n^{1:t'})$, $t' < t - \Delta$ and ii) we only used $X_n^{t'}, D_n^{t'}, t' = 1, \ldots t$, for training.

CP algorithms are applied on top of given point-prediction models, where we will use $\mathcal{X}$ to denote the cross-sectional sample space of any $X_t$. In our case, the underlying model is $f$, which predicts the expected user behaviour at time $t + \Delta$ given the user history up to time $t$, i.e. $\hat{X}_n^{t+\Delta} := f(D_n^{1:t}, X_n^{1:t}, Z_n) \approx \mathbb{E}[X_n^{t+\Delta} \mid D_n^{1:t}, X_n^{1:t}, Z_n, \text{do}(D^{1:t}, d^{t+1}, 0^{t+2:t+\Delta})]$ for a chosen implicit $d^{t+1}$ and with $0^{t+2:t+\Delta}$ denoting the default action 0 being taken at $t + 2, \ldots, t + \Delta$.

**Setup 1.** Calibration and test scores, $\{S_n = |X_n^{t+\Delta} - f(D_n^{1:t}, X_n^{1:t}, Z_n)|\}_{n=1}^N$ and $S_{N+1} = |X_{N+1}^{t+\Delta} - f(D_{N+1}^{1:t}, X_{N+1}^{1:t}, Z_n)|$ are exchangeable, i.e. $\text{Prob}(S_1, \ldots, S_N, 1) = \text{Prob}(S_{\sigma(1)}, \ldots, S_{\sigma(N+1)})$ where $\sigma$ is any permutation of $\{1, \ldots, N + 1\}$. The Quantile Lemma, e.g. Lemma 1 of [44], implies the prediction interval

$$C = \{x \in \mathcal{X}, |x - \hat{X}_{N+1}^{t+\Delta}| \leq \hat{Q}_\alpha\} \tag{8}$$

$$\hat{Q}_\alpha = \inf_q \left\{ \sum_{n=1}^N \mathbf{1}(|X_n^{t+\Delta} - \hat{X}^{t+\Delta}| \leq q) \geq n_\alpha \right\}, \quad n_\alpha = \lceil (1 + N)(1 - \alpha) \rceil$$

is *valid* in the sense it obeys

$$\text{Prob}\left(X_{N+1}^{t+\Delta} \in C\right) \geq 1 - \alpha, \tag{9}$$

where the probability is over the calibration and test samples.

**Setup 2.** The prediction intervals defined in (8) may not be valid, i.e. (9) may not hold, because the calibration and test samples, $S_n^{t'} = |X_n^{t'+\Delta} - f(D_n^{1:t'}, X_n^{1:t'}, Z_n)|$ with $t' \leq t$ and $t' > t$, $n = 1, \ldots, N$, are not exchangeable. Theorem 2 provides a bound on the coverage gap, i.e. a measure of violation of (9), under the assumption that the distribution shift is controlled by a perturbation parameter, $\epsilon > 0$.

**Theorem 2** *Assume we have $N$ calibration samples,*

$$S_n^{t'} = |X_n^{t'} - f(D_n^{1:t_n}, X_n^{1:t_n}, Z_n)|, \quad t' = t_n + \Delta < t, \quad n = 1, \ldots, N \tag{10}$$

*where $t_n$ is the time user $n$ experienced the last intervention before $t$. Assume there exists $\epsilon > 0$ such that, for all $n$,*

$$p_T(S_n^{t+\Delta}) = (1 - \epsilon)p_C(S_n^{t+\Delta}) + \epsilon p_\delta(S_n^{t+\Delta}), \tag{11}$$

*where $p_T$ and $p_C$ are the (unknown) densities of the test and calibration distributions, $p_\delta$ is a bounded arbitrary shift density, and $p_{min} = \min_{n=1,\ldots,N} p_C(S_n^{t'}) > 0$. Then,*

$$\text{Prob}\left(X_{N+1}^{t+\Delta} \in C\right) \geq 1 - \alpha - \frac{1}{p_{min}} \frac{\epsilon}{1 - \epsilon}. \tag{12}$$

In the proof, we use the likelihood-ratio-weighting approach of [44] to obtain the empirical test distribution from the calibration samples and bound its quantile from below. The statement follows from standard inequalities on the mean and variance of the $\epsilon$-perturbed distribution. We prefer this approach to conformal prediction adaptive models for time series, e.g. [20] or [6], for two reasons: i) adaptive schemes have asymptotic coverage guarantees that can not be used to estimate the uncertainty on a single time step and ii) optimized density estimates are a byproduct of our prediction model.

## 4  Further Related Work

[9] leverages Bayesian structural time-series models to estimate causal effects, and motivates our exponential decay model in Eq. (3). Unlike [9], who focus on single interventions, our model explicitly addresses the complexity arising from sequential interventions, taking a more detailed perspective on the dynamic interplay of treatments over time. Several approaches for conformal prediction in causal inference have emerged in recent years [e.g., 44, 28], including matrix completion [21] and synthetic controls [12]. Our focus has not been on individual nor average treatment effects, but directly on expected potential outcomes, similarly to the work on causal matrix completion [7, 3, 4, 42]. Unlike the matrix completion literature, we focused on prediction problems, out-of-sample for both units and time steps. Moreover, the matrix completion literature is usually framed in terms of *marginal* expectations $\mathbb{E}[X_n^t(d)]$, as opposed to conditional expectations (notice that [4] considers covariates, but those are included as part of the generative model). Marginal models have some advantages when modeling multiple-step-ahead effects [18], but they also involve complex computational considerations that we leave for future work. Compositionality based on additivity instead of the factorization trick is discussed by e.g. [29], but additivity lacks the connection to an explicit principle for universal function approximation. Finally, there is a rich literature on confounding adjustment where exogeneity of $D_n^t$ cannot be assumed, see [22] for a textbook treatment. We can rely on standard approaches of sequential ignorability [36] to justify our method in the absence of randomization.

## 5  Experiments

We run a number of synthetic and semi-synthetic experiments to evaluate the performance of the CSI-VAE approach. In this section, we summarize our experimental set-up and main results. The code for reproducing all results and figures is available online[2]. In Appendix C, we provide a detailed description of the datasets and models. In Appendix D, we present further analysis and more results. Finally, in Appendix E, we present an illustration of uncertainty quantification results.

---

[2]The code is available at https://github.com/jialin-yu/CSI-VAE

Table 1: Main experimental results, averaged mean squared root error over five different seeds.

| Model | Full Synthetic | | | | | Semi-Synthetic Spotify | | | | |
|---|---|---|---|---|---|---|---|---|---|---|
| | T+1 | T+2 | T+3 | T+4 | T+5 | T+1 | T+2 | T+3 | T+4 | T+5 |
| CSI-VAE-1 | 36.53 | 41.46 | 41.73 | 41.12 | 41.32 | 68.23 | 82.94 | 83.53 | 81.97 | 79.63 |
| CSI-VAE-2 | 97.80 | 118.25 | 117.79 | 127.25 | 135.03 | 253.85 | 312.53 | 305.08 | 303.68 | 302.83 |
| CSI-VAE-3 | 138.78 | 164.02 | 141.71 | 132.59 | 125.55 | 757.94 | 937.07 | 800.55 | 704.66 | 634.72 |
| GRU-0 | 229.72 | 269.66 | 220.95 | 208.30 | 188.43 | 215.42 | 260.65 | 193.41 | 137.20 | 117.06 |
| GRU-1 | 230.76 | 270.83 | 220.93 | 208.33 | 184.92 | 223.61 | 269.69 | 205.91 | 141.53 | 126.36 |
| GRU-2 | 93.73 | 101.03 | 118.01 | 88.53 | 132.28 | 154.18 | 187.42 | 177.96 | 133.36 | 127.58 |
| LSTM | 114.71 | 126.65 | 137.12 | 105.22 | 137.19 | 130.35 | 156.02 | 133.28 | 94.35 | 85.92 |
| Transformer | 111.66 | 122.08 | 150.57 | 175.84 | 87.89 | 133.42 | 157.66 | 154.61 | 164.70 | 158.03 |

Table 2: P-values from two-sample t-tests against the null hypothesis that models perform no better on average than its counterpart. Significance at $0.05$ indicated with one asterisk, $0.001$ with two.

| Model | Full Synthetic | | | | | | | Semi-Synthetic Spotify | | | | | | |
|---|---|---|---|---|---|---|---|---|---|---|---|---|---|---|
| | CSI-VAE-2 | CSI-VAE-3 | GRU-0 | GRU-1 | GRU-2 | LSTM | Transformer | CSI-VAE-2 | CSI-VAE-3 | GRU-0 | GRU-1 | GRU-2 | LSTM | Transformer |
| CSI-VAE-1 | < 0.001** | < 0.001** | < 0.001** | < 0.001** | < 0.001** | < 0.001** | < 0.001** | < 0.001** | < 0.001** | < 0.001** | < 0.001** | < 0.001** | < 0.001** | < 0.001** |
| CSI-VAE-2 | \ | 0.057 | < 0.001** | < 0.001** | 0.249 | 0.627 | 0.510 | \ | < 0.001** | 0.058 | 0.081 | < 0.05* | < 0.05* | < 0.05* |
| CSI-VAE-3 | \ | \ | < 0.001** | < 0.001** | < 0.05* | 0.228 | 0.544 | \ | \ | < 0.001** | < 0.001** | < 0.001** | < 0.001** | < 0.001** |
| GRU-0 | \ | \ | \ | 0.990 | < 0.001** | < 0.001** | < 0.001** | \ | \ | \ | 0.705 | 0.129 | < 0.05* | 0.109 |
| GRU-1 | \ | \ | \ | \ | < 0.001** | < 0.001** | < 0.001** | \ | \ | \ | \ | 0.061 | < 0.001** | 0.052 |
| GRU-2 | \ | \ | \ | \ | \ | 0.195 | 0.205 | \ | \ | \ | \ | \ | < 0.05* | 0.870 |
| LSTM | \ | \ | \ | \ | \ | \ | 0.758 | \ | \ | \ | \ | \ | \ | < 0.05* |

**Datasets and oracular simulators.** The first step to assess intervention predictions is to build a set of proper ground truth test beds, where we can control different levels of combinations of interventions. We build two sets of oracular simulators. (1) The **Fully-synthetic** simulator is constructed primarily based on random models following our parameterization in Section 2. (2) The **Semi-synthetic** simulator is constructed based on a real-world dataset from Spotify[3] which aims to predict skip behavior of users based on their past interaction history. See details in Appendix C. For each type of simulator, we generate 5 different versions with different random seeds. From simulator (1), we construct a simulated dataset of size $50,000$, containing 5 different interventions happening to the units at any time after an initial burn-in period of $T_0 = 10$, although only maximum 3 of these can be observed for any given unit. We set $r = 5$ as the true dimensionality of the model. For simulator (2), we construct simulated datasets of size $3,000$, again containing 5 different interventions, an initial period $T_0 = 25$, a maximum of 3 different interventions per unit, and $r = 10$. The task is to predict outcomes for interventions not applied yet within any given unit (i.e., at least from the 2 options left). In simulator (2), parameters $\phi$ and $\beta$ are learned from real-world data. Interventions are artificial, but inspired by the process of showing different proportions of track types to an user in a Spotify-like environment. For both setups, we use a data ratio of $0.7, 0.1, 0.2$ for training, validation and test, respectively. We report the final results in Table 1 with another constructed holdout set ($5,000$ points for the fully-synthetic case, and $1,000$ for the semi-synthetic one).

**Compared models.** We implemented three variations: (1) **CSI-VAE-1** follows exactly our setup in Section 2; (2) **CSI-VAE-2** can be considered as an ablation study, which relaxes the product form of Eq. (1) and replace it with a black-box MLP applied directly to $(\phi_n^t, \beta_n, \psi_n^t)$ (and hence may not guarantee identifiability); (3) **CSI-VAE-3** is another ablation study, where the equation for $\psi$ (Eq. (2)) is replaced by a black-box function taking the sequence of actions $D$ as a standard time-series. We compare our model against: (1) **GRU-0**, a black-box gated recurrent unit (GRU, [13]) composed with a MLP, using only the past history of $X_n^{1:t}$ and $Z_n$; (2) **GRU-1**, another GRU composed with a MLP that takes into account also the latest intervention $D_n^t$; and (3) **GRU-2**, which uses not only $D_n^t$ but also the entire past history $D_n^{1:t}$ just like CSI-VAE. In general, GRU-2 can be considered as a very strong and generic black-box baseline model. In addition, we conduct experiments comparing other popular black-box baseline models: (4) **LSTM**, [23]; and (5) **Transformer**, [45]. For those, we use the same input setup as in the case of the GRU-2 model.

**Results.** Each experiment was repeated 5 times, using Adam [26] at a learning rate of 0.01, with 50 epochs in the fully-synthetic case and 100 for the semi-synthetic, which was enough for convergence. We selected the best iteration point based on a small holdout set. The main results are presented in Tables 1 and 2, which show the superiority of our model against strong baselines. We also observed

---

[3] https://open.spotify.com/

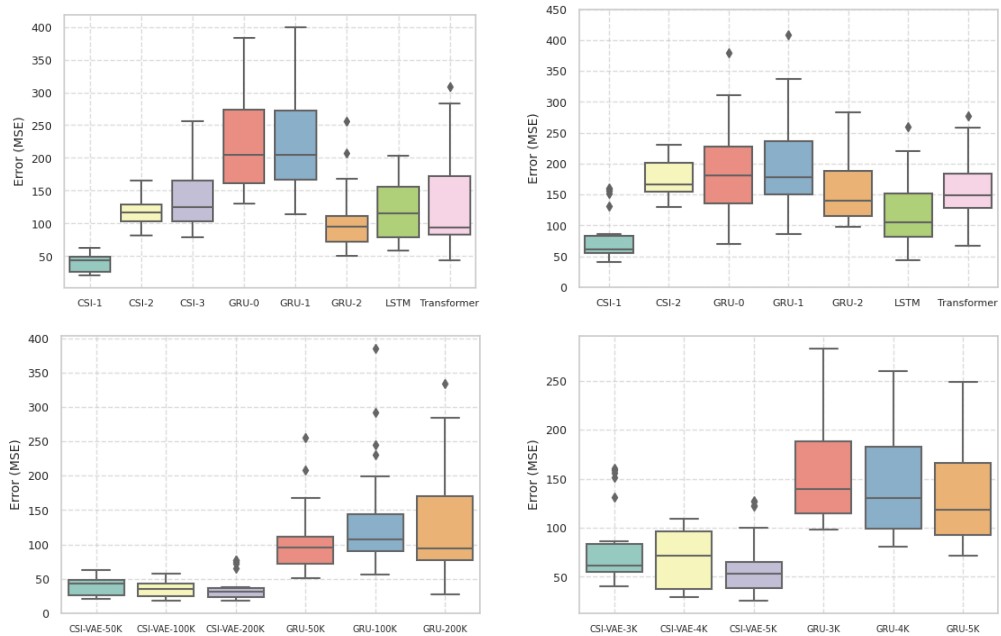

Figure 2: **Top**: 5-run evaluation of test mean squared error on the fully-synthetic (left) and semi-synthetic cases (case). CSI-3 was removed on the right due to very high errors. **Bottom**: how errors change as training sizes are increased, CSI-1 vs. GRU-2 (left: fully-synthetic, right: semi-synthetic).

that the identifiability results and compositional interactions of intervention effects are both critical, as evidenced by the drop in performance for CSI-VAE 2 and 3 in Table 1.

In Appendix D, we provide the following further experiments: (1) different choices of $r$ (summary: using $r$ less than the true value gracefully underfits, while there is evidence of some overfitting for choices of $r$ which overshoot the true value — mitigated by regularization); (2) different sizes of the training data (summary: even with more data provided, our model consistently outperforms the black-box models by a significant margin; we show that a generic black-box cannot solve this problem by simply feeding in more data); and (3) a demonstration of conformal prediction that allow us to better calibrate the predictive coverage compared to vanilla model-based prediction.

## 6 Conclusion

We introduced an approach for predictions of sequences under hypothetically controlled actions, with a careful accounting of when extrapolation to unseen sequences of controls is warranted.

**Findings.** Embeddings are important given sparse categorical sequential data [45]. However, large combinatorial interventional problems benefit from models that carefully lay down conditions for the identification of such embeddings. Naïve sequential models, however flexible, cannot fully do the heavy lifting of generalizing in the absence of structure. Information has to come from somewhere.

**Limitations.** Unlike the traditional synthetic control literature [1], we assume a model for time effects based on autoregression and truncated or parametric time/intervention interactions. While it is possible to empirically evaluate the predictive abilities of the model using a validation sample, high-stakes applications (such as major interventions to counteract the effect of a pandemic) should take into consideration that uncontrolled distribution shifts may take place, and careful modeling of such shifts should be added to any analytical pipeline to avoid damaging societal implications.

**Future work.** Besides allowing for an explicit parameterization of drifts, accounting for unmeasured confounding that may take place among past actions and states is necessary to increase the applicability of the method. Moreover, when each action level $d$ is itself a combination of cross-sectional actions, causal energy-based models such as [8] can be combined with the compositional factorization idea to also generalize to previously unseen cross-sectional combinations of actions.

## Acknowledgements

We thank Zhenwen Dai and Georges Dupret from Spotify for numerous helpful comments and suggestions. JY and RS were partially funded by the EPSRC Open Fellowship *The Causal Continuum: Transforming Modelling and Computation in Causal Inference*, EP/W024330/1. RS acknowledges support of the UKRI AI programme, and the Engineering and Physical Sciences Research Council, for *CHAI – EPSRC AI Hub for Causality in Healthcare AI with Real Data* (grant number EP/Y028856/1). The authors would also like to thank the four anonymous reviewers for fruitful suggestions on how to improve the presentation of this paper.

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

# Appendix

## A General Background

Much of traditional causal inference aims at estimating the outcome of a single-shot treatment that is not intertwined within a sequence with uncontrolled intermediate outcomes [24]. However, many causal questions involve sequential treatments. For example, we might be interested in estimating the causal effects of medical treatments, lifestyle habits, employment status, marital status, occupation exposures, etc. Because these treatments may take different values for a single individual over time, they are often refereed to as time-varying treatments [22].

In many designs, it is assumed that a time-fixed treatment applied to individuals in the population happens at the same time (i.e. there is a shared global clock for everyone). A more relaxed setting (and more practical, as described in this paper) is that (categorical) interventions can happen to different individuals at different time steps, potentially resulting in no pair of identically distributed time-series. The purpose now is to estimate the causal effect or to predict expected outcomes under time-varying treatments as a time-series, rather than estimating/imputing missing potential outcome snapshots from a fixed time interval that has taken place in the past.

One typical way of designing experiments under a binary treatment (0 or 1) [35, 36] of time-varying treatments is to have $\bar{d}_0 = (0, 0, \cdots, 0)$ ("never treated") and $\bar{d}_1 = (1, 1, \cdots, 1)$ ("always treated") as the two options in the design space. This allows us to answer what the average expected outcome contrasted between these two options. A fine-grained estimation at every time step will require further levels of contrast for other regimes, such as $d = (0, 1, \cdots, 1)$ and $d = (1, 0, \cdots, 0)$. However, the number of combinations grows exponentially on $T$, the number of time steps, even when only binary treatments are at the stake.

In this paper, we did not consider identification strategies that may be necessary when confounding is an issue. Conditional ignorability holds when we are able to block unmeasured confounders using observable confounders only. Similarly, in a time-varying setting, we achieve sequential conditional ignorability by conditioning on the time-varying covariates, including past actions ([22], and Chapter 4 of [33]). In our manuscript, we will have little to say about the uses of conditional sequential ignorability and other adjustment techniques for observational studies, as these ideas are well-understood and can be applied on top of our main results. Hence, we decided to focus on treatments that act as-if randomized so that we can explore in more detail our contributions.

## B Proofs

**Proposition 1** *Let* **(i)** $f(n, d^{1:T}, x^{1:T}, z) := f_d^\mathsf{T}(d^{1:T}) f_{nxz}(n, x^{1:T}, z)$, *where function sequences* $f_d(d^{1:T})$ *and* $f_{nxz}(n, x^{1:T}, z)$ *are defined for all* $T \in \mathbb{N}^+$ *and have codomain* $\mathbb{R}^{r_g}$; **(ii)** $f_{dl}(d^{1:T})$ *be given as in Eq. (6) with* $r_{h_l} = 1$; **(iii)** *the* $l$-*th entry of* $f_{nxz}(n, x^{1:T}, z)$ *be given by* $f_{nxzl}(n, x^{1:T}, z) := u_l(n)^\mathsf{T} v_l(x^{1:T}, z)$, *where both* $u_l(\cdot)$ *and* $v_l(\cdot, \cdot)$ *have codomain* $\mathbb{R}^{r_{s'}}$; **(iv)** $v_l(x^{1:T}, z)$ *be defined analogously to* $f_{dl}(d^{1:t})$ *as in Eq. (6), carrying out the fixed-size* $z$ *as an extra argument. Then there exists some integer* $r$ *and three functions* $a, b$ *and* $c$ *with codomain* $\mathbb{R}^r$ *so that* $f(n, d^{1:T}, x^{1:T}, z) = \sum_{l=1}^r a_l(x^{1:T}, z) \times b_l(n) \times c_l(d^{1:T}, T)$. *Moreover* $c_l(d^{1:T}, T) = \prod_{t=1}^T m_l(d^t, t, T)$ *for some function* $m_l : \mathbb{N}^3 \to \mathbb{R}$.

**Proof.** The proof is tedious and the result intuitive, but they help to formalize the main motivation for Eqs. (1) and (3).

$$
\begin{aligned}
f_d^\mathsf{T}(d^{1:T}) f_{nxz}(n, x^{1:T}, z) &= \sum_{l'=1}^{r_s} f_{dl'}(d^{1:T}) f_{nxzl'}(n, x^{1:T}, z) \\
&= \sum_{l'=1}^{r_s} f_{dl'}(d^{1:T}) \sum_{l''}^{r_{s'}} u_{l'l''}(n) v_{l'l''}(x^{1:T}, z) \\
&= \sum_{l''=1}^{r_{s'}} \sum_{l'=1}^{r_s} u_{l'l''}(n) v_{l'l''}(x^{1:T}, z) f_{dl'}(d^{1:T}) \\
&= \sum_{l=1}^r a_l(x^{1:T}, z) \times b_l(n) \times c_l(d^{1:T}),
\end{aligned}
$$

where $r = r_s \times r_{s'}$ and $a_l(x^{1:T}, z) := v_{l'l''}(x^{1:T}, z)$ for $l = l' \times l''$; $b_l(n) := u_{l'l''}(n)$ for $l = l' \times l''$; and $c_l(d^{1:T}) = f_{dl'}(d^{1:T})$ for $l' = \lceil l/r_{s'} \rceil$, where $\lceil \cdot \rceil$ is the ceiling function.

Finally, each $c_l(d^{1:T})$ is by definition some $f_{dl'}(d^{1:T})$. From Eq. (6) and the assumptions, $f_{dl'}(d^{1:T}) = g_{l'}(d^{1:T}, T, T)$, and $g_{l'}(d^{1:t}, t', t) := h_{l'}(g_l(d^{1:t'-1}, t'-1, t), d^{t'}, t', t)$. For $t' > 0$ and $d > 0$, we have that $h'_l$ is given by $h_{l'_1}(g_{l'}(d^{1:t}, t'-1, t))h_{l'_2}(d^t, t', t)$. For $t' = 0$, $h_{l'}(v, d, t', t) \equiv 1$, and for $t' > 0$ and $d = 0$, $h_{l'}(v, d, t', t) \equiv g_{l'}(d^{1:t'-1}, t'-1, t)$. By recursive application, this implies $f_{dl'}(d^{1:T}) = \prod_{t'=1:d^{t'}>0}^T h_{l'_2}(d^{t'}, t', T)$ and the result follows. $\square$

**Proposition 2**    *Given a fixed value of $T$, suppose we have a measurable function $f : \bigcup_{t=2}^T \mathcal{X}^{1:t-1} \times \bigcup_{t=1}^T \mathcal{D}^t \times \mathcal{Z} \to \mathbb{R}^d$, $t \le T$, where $\mathcal{X}^{1:t-1}$ and $\mathcal{Z}$ are compact, $\mathcal{D}^t$ is finite. Let $\mathbf{x} \in \bigcup_{t=2}^T \mathcal{X}^{1:t-1}$ and let $\mathbf{d} \in \bigcup_{t=1}^T \mathcal{D}^t$. For $d = 1, \cdots, d_{max}$, let $m_d : \bigcup_{t=1}^T \mathcal{D}^t \to \{1, \cdots, T\}$ with $m_d(\mathbf{d}) = t$ if $\exists t$ s.t. $\mathbf{d}_t = d$ else $0$. Then, for any $\epsilon > 0$, there exist $r = O(T^{d_{max}+1})$ and measurable functions $\phi_l : \mathcal{X}^{1:t-1} \times \mathcal{Z} \to \mathbb{R}^d$ and $\psi_{ld'} : \mathcal{D}^t \to \mathbb{R}$, and real numbers $\beta_l$, such that*

$$sup_{\mathbf{x} \in \bigcup_{t=2}^T \mathcal{X}^{1:t-1}, \mathbf{d} \in \bigcup_{t=1}^T \mathcal{D}^t, z \in \mathcal{Z}} \left| f(\mathbf{x}, \mathbf{d}, z) - \sum_{l=1}^r \phi_l(\mathbf{x}, z) \times \beta_l \times \prod_{d=1}^{d_{max}} \psi_{ld'}(m_{d'}(\mathbf{d})) \right| \le \epsilon. \quad (13)$$

*Proof.* First, let

$$\hat{f}(\mathbf{x}, \mathbf{d}, z) := \sum_{l=1}^r \phi_l(\mathbf{x}, z) \times \beta_l \times \prod_{d=1}^{d_{max}} \psi_{ld'}(m_{d'}(\mathbf{d})).$$

Notice that we can construct an equivalent definition of $D^t$ for some fixed $t$ as follows. We for now call the new object $\bar{D}^t$:

$$\bar{\mathcal{D}}^t = \left\{ \mathbf{v} \in \mathbb{R}^{d_{max}} \middle| v_d \in \{0, \cdots, t\} \forall d; \ v_{d'} \ne v_d \text{ whenever } d' \ne d \text{ and } v_d \ne 0 \right\}. \quad (14)$$

Note that there is a one-to-one correspondence between $\bar{\mathcal{D}}^t$ and $\mathcal{D}^t$, and denote the bijection connecting them by $g : \bar{\mathcal{D}}^t \to \mathcal{D}^t$. Since both sets are finite, we can approximate $f$ with $\hat{f}$, if and only if we can approximate $f \circ i \times g \times i$ by $\hat{f} \circ i \times g \times i$. So, without loss of generality, just consider $\bar{\mathcal{D}}^t$ as $\mathcal{D}^t$.

The set of all sequences of treatments is the union of $\mathcal{D}^t$ over $t = 1, ..., T$, for some known $T$:

$$\mathcal{D}^{\mathcal{T}} = \bigcup_{t=1}^T \mathcal{D}^t \quad \text{known } T. \quad (15)$$

Note that

$$|\mathcal{D}^t| < (t+1)^{d_{max}} \quad (16)$$

$$|\mathcal{D}^{\mathcal{T}}| < \sum_{t=1}^T (t+1)^{d_{max}} \quad (17)$$

therefore the size of $\mathcal{D}^{\mathcal{T}}$ is polynomial in $T$.

Fixing the ground truth function $f$ at $D \in \mathcal{D}^{\mathcal{T}}$ gives us a new function $\bigcup_{t=1}^T \mathcal{X}^{1:t-1} \times \mathcal{Z} \to \mathcal{X}$. Analogously, fixing $\hat{f}$ at some $D \in \mathcal{D}^{\mathcal{T}}$ also gives a new function in the same function space. Since $\mathcal{D}^{\mathcal{T}}$ is finite, we can enumerate its elements. Thus, for every $D^{(i)} \in \mathcal{D}^{\mathcal{T}}$, $i = 1, \cdots, |\mathcal{D}^{\mathcal{T}}|$, write down $\hat{f}$ fixed at $D^{(i)}$ as:

$$\hat{f}(X^{1:t-1}, D^{(i)}, Z) = \sum_{l=1}^r \phi_l(X^{1:t-1}, Z)\beta_l \psi_{l1}(D_1^{(i)})\psi_{l2}(D_2^{(i)}) \cdots \psi_{ld_{max}}(D_{d_{max}}^{(i)}). \quad (18)$$

We can then express the vector field as:

$$\begin{pmatrix} \hat{f}(\cdot, D^{(1)}, \cdot) \\ \vdots \\ \hat{f}(\cdot, D^{(|\mathcal{D}^{\mathcal{T}}|)}, \cdot) \end{pmatrix} = \underbrace{\begin{pmatrix} \beta_1 \prod_{d=1}^{d_{max}} \psi_{11}(D_1^{(1)}) & \cdots & \beta_r \prod_{d=1}^{d_{max}} \psi_{r1}(D_1^{(1)}) \\ \vdots & \vdots & \vdots \\ \beta_1 \prod_{d=1}^{d_{max}} \psi_{11}(D_1^{(|\mathcal{D}^{\mathcal{T}}|)}) & \cdots & \beta_r \prod_{d=1}^{d_{max}} \psi_{r1}(D_1^{(|\mathcal{D}^{\mathcal{T}}|)}) \end{pmatrix}}_W \begin{pmatrix} \phi_1(\cdot, \cdot) \\ \vdots \\ \phi_r(\cdot, \cdot) \end{pmatrix}$$

$$(19)$$

By choosing $r = |\mathcal{D}^{\mathcal{T}}|$ and appropriate values for $\beta_l$ and $\psi_{ld}(D^{(i)})$, we can make $W$ invertible.

Now, consider the vector field given by the ground truth function fixed at all values of $D \in \mathcal{D}^{\mathcal{T}}$:

$$\begin{pmatrix} f(\cdot, D^{(1)}, \cdot) \\ \vdots \\ f(\cdot, D^{(|\mathcal{D}^{\mathcal{T}}|)}, \cdot) \end{pmatrix}, \tag{20}$$

and consider its transformation under $W^{-1}$:

$$W^{-1} \begin{pmatrix} f(\cdot, D^{(1)}, \cdot) \\ \vdots \\ f(\cdot, D^{(|\mathcal{D}^{\mathcal{T}}|)}, \cdot) \end{pmatrix}. \tag{21}$$

Since recurrent neural networks are universal approximators [40], for every $\delta > 0$ we can choose $\phi_1, \cdots, \phi_r$ such that

$$\sup_{\mathbf{x} \in \bigcup_{t=2}^{T} \mathcal{X}^{1:t-1}, \, z \in \mathcal{Z}} \left\| \begin{pmatrix} \phi_1(\mathbf{x}, z) \\ \vdots \\ \phi_r(\mathbf{x}, z) \end{pmatrix} - W^{-1} \begin{pmatrix} f(\mathbf{x}, D^{(1)}, z) \\ \vdots \\ f(\mathbf{x}, D^{(|\mathcal{D}^{\mathcal{T}}|)}, z) \end{pmatrix} \right\|_2^2 \leq \delta. \tag{22}$$

Since $W$ is a finite matrix, its operator norm is well-defined. Then

$$\sup_{\mathbf{x} \in \bigcup_{t=2}^{T} \mathcal{X}^{1:t-1}, \, z \in \mathcal{Z}} \left\| W \begin{pmatrix} \phi_1(\mathbf{x}, z) \\ \vdots \\ \phi_r(\mathbf{x}, z) \end{pmatrix} - \begin{pmatrix} f(\mathbf{x}, D^{(1)}, z) \\ \vdots \\ f(\mathbf{x}, D^{(|\mathcal{D}^{\mathcal{T}}|)}, z) \end{pmatrix} \right\|_2^2 \tag{23}$$

$$\leq \sup_{\mathbf{x} \in \bigcup_{t=2}^{T} \mathcal{X}^{1:t-1}, \, z \in \mathcal{Z}} \|W\|_{op}^2 \left\| \begin{pmatrix} \phi_1(\mathbf{x}, z) \\ \vdots \\ \phi_r(\mathbf{x}, z) \end{pmatrix} - W^{-1} \begin{pmatrix} f(\mathbf{x}, D^{(1)}, z) \\ \vdots \\ f(\mathbf{x}, D^{(|\mathcal{D}^{\mathcal{T}}|)}, z) \end{pmatrix} \right\|_2^2 \tag{24}$$

$$\leq \|W\|_{op}^2 \sup_{\mathbf{x} \in \bigcup_{t=2}^{T} \mathcal{X}^{1:t-1}, \, z \in \mathcal{Z}} \left\| \begin{pmatrix} \phi_1(\mathbf{x}, z) \\ \vdots \\ \phi_r(\mathbf{x}, z) \end{pmatrix} - W^{-1} \begin{pmatrix} f(\mathbf{x}, D^{(1)}, z) \\ \vdots \\ f(\mathbf{x}, D^{(|\mathcal{D}^{\mathcal{T}}|)}, z) \end{pmatrix} \right\|_2 \tag{25}$$

$$\leq \|W\|_{op}^2 \delta. \tag{26}$$

However,

$$\sup_{\mathbf{x} \in \bigcup_{t=2}^{T} \mathcal{X}^{1:t-1}, \, \mathbf{d} \in \mathcal{D}^{\mathcal{T}} \, z \in \mathcal{Z}} |f(\mathbf{x}, \mathbf{d}, z) - \hat{f}(\mathbf{x}, \mathbf{d}, z)| \tag{27}$$

$$= \left( \sup_{\mathbf{x} \in \bigcup_{t=2}^{T} \mathcal{X}^{1:t-1}, \, z \in \mathcal{Z}} \max_{\mathbf{d} \in \mathcal{D}^{\mathcal{T}}} |f(\mathbf{x}, \mathbf{d}, z) - \hat{f}(\mathbf{x}, \mathbf{d}, z)|^2 \right)^{1/2} \tag{28}$$

$$\leq \left( \sup_{\mathbf{x} \in \bigcup_{t=2}^{T} \mathcal{X}^{1:t-1}, \, z \in \mathcal{Z}} \sum_{\mathbf{d} \in \mathcal{D}^{\mathcal{T}}} |f(\mathbf{x}, \mathbf{d}, z) - \hat{f}(\mathbf{x}, \mathbf{d}, z)|^2 \right)^{1/2} \tag{29}$$

$$= \left( \sup_{\mathbf{x} \in \bigcup_{t=2}^{T} \mathcal{X}^{1:t-1}, \, z \in \mathcal{Z}} \left\| W \begin{pmatrix} \phi_1(\mathbf{x}, z) \\ \vdots \\ \phi_r(\mathbf{x}, z) \end{pmatrix} - \begin{pmatrix} f(\mathbf{x}, D^{(1)}, z) \\ \vdots \\ f(\mathbf{x}, D^{(|\mathcal{D}^{\mathcal{T}}|)}, z) \end{pmatrix} \right\|_2^2 \right)^{1/2} \tag{30}$$

$$\leq \|W\|_{op}^2 \delta. \tag{31}$$

Therefore, choosing $\delta = \epsilon / \|W\|_{op}^2$ gives us the desired result.

$\square$

**Proposition 3** *Let $b_n$ be a $(T_{0_n} - 1) \times 1$ vector with entries $\mathbb{E}[X_n^{t+1} \mid x_n^{1:t}, z_n, \mathrm{do}(d_n^{1:t+1})]$. Under Assumptions 1 and 2, $\beta_n$ is identifiable from $A_n$ and $b_n$.*

**Proof.** Let $b_n$ be a $(T_{0_n} - 1) \times 1$ vector where each entry $t$ is given by $\mathbb{E}[X_n^{t+1} \mid x_n^{1:t}, z_n, \mathrm{do}(0^{1:t+1})]$, where $0^{1:t+1}$ denotes all actions kept at the default level of 0 from time 1 to time $t + 1$. We know from standard least-squares theory, and the given assumptions, that as long as the number $T_0 \geq r$ of observations is larger than or equal to the dimension $r$ and the matrix is full rank, that the system is (over-)determined and $\beta_n = A_n^+ b$. $\square$

**Theorem 1** *Assume we have a dataset of $N$ units and $T$ time points $(d_n^{1:T}, x_n^{1:T}, z_n)$ generated by a model partially specified by Eq. (1). Assume also knowledge of the conditional expectations $\mathbb{E}[X_n^t \mid x_n^{t-1}, z_n, \mathrm{do}(d_n^{1:t})]$ and basis functions $\phi_l(\cdot, \cdot)$ for all $l = 1, 2, \ldots, r$ and all $1 \leq n \leq N$ and $1 \leq t \leq T$. Then, under Assumptions 1-4 applied to all individuals and all intervention levels $d$ that appear in our dataset, we have that all $\beta_n$ and all $\psi_l(d, \cdot, \cdot)$ are identifiable.*

**Proof.** By Assumptions 1 and 2, and Proposition 3, all $\beta_n$ are identifiable.

For the next step, let

$$\eta_{nl}^t := \beta_{nl} \times \prod_{t'=1}^{t} \psi_l(d_n^{t'}, t', t),$$

and

$$\alpha_{dl}^{\Delta_t} := \prod_{t'=1}^{\Delta_t} \psi_l(d, 1, t').$$

Consider now the first time $t$ where an intervention level $d > 0$ is assigned to someone in the entire dataset. By Assumption 3, there exists a dataset $\mathcal{N}_d$ with the given properties where, for all $n \in \mathcal{N}_d$, we have $\eta_{nl}^t = \beta_{nl} \times \prod_{t'=1}^{t} \psi_l(d_n^{t'}, t', t) = \beta_{nl}$, since all interventions prior to $t$ have been at level 0.

Let $b_{dt'}$ be a $N_d \times 1$ vector where each entry $i$ is given by $\mathbb{E}[X_{n_i}^{t+t'} \mid x_{n_i}^{1:t+t'-1}, z_{n_i}, \mathrm{do}(d_{n_i}^{1:t+t'})]$, for $t' = 1, 2, \ldots, k_d - 1$. Consider the time-bounded case with free parameters $w_{dl,1}$, $l = 1, 2, \ldots, r$. Given Assumption 4, we can solve for $\eta_n^{t+1} = A_{d_1}^+ b_{d_1}$, the vector with entries $\eta_{nl}^{t+1}$.

As $\eta_{nl}^{t+1} = \eta_{nl}^t \times \psi_l(d, t+1, t+1) = \eta_{nl}^{t+1} \times w_{dl,1}$, with non-zero $\eta_{nl}$ by Assumption 3, this also identifies $w_{dl,1}$. Also by Assumption 3, at time point $t+2$ we have that $\eta_{nl}^{t+1} = \eta_{nl}^t \times \psi_l(d, t+1, t+2)$ for all units in $\mathcal{N}_d$, which by analogy to the previous paragraph, identifies $\psi_l(d, t+1, t+2)$ and consequently $w_{dl,2}$. As this goes all the way to $t + k_d - 1$, this identifies all of $\psi_l(d, \cdots, \cdot)$.

If intervention $d$ level is the time-unbounded model of Eq. (3), by Assumption 3, we also identify $\alpha_d^1$, $\alpha_d^2$ and $\alpha_d^3$, as they are defined by functions $\psi_l(d, t+1, t+t') = \psi_l(d, 1, t')$ for $t' = 1, 2, 3$ that we established as identified by the reasoning in the previous paragraph. As $\sigma(w_{1dl}) \times w_{2dl} - \sigma(w_{1dl})^3 \times w_{2dl} = \alpha_{dl}^1 - \alpha_{dl}^3$ and

$$\frac{\sigma(w_{1dl}) \times w_{2dl} - \sigma(w_{1dl})^2 \times w_{2dl}}{\sigma(w_{1dl})^2 \times w_{2dl} - \sigma(w_{1dl})^3 \times w_{2dl}} = \frac{\alpha_{dl}^1 - \alpha_{dl}^2}{\alpha_{dl}^2 - \alpha_{dl}^3},$$

solving for the above results in

$$w_{1dl} = \sigma^{-1}\left(\frac{\alpha_{dl}^2 - \alpha_{dl}^3}{\alpha_{dl}^1 - \alpha_{dl}^2}\right),$$

$$w_{2dl} = \frac{\alpha_{dl}^1 - \alpha_{dl}^3}{\sigma(w_{1dl}) - \sigma(w_{1dl})^3},$$

$$w_{3dl} = \alpha_{dl}^1 - \sigma(w_{1dl}) \times w_{2dl}.$$

So far we have established identification of the $\psi(d, \cdot, \cdot)$ functions for the first intervention level $d$ that appears in the dataset (the value $d$ is not unique, as it is possible to assign different intervention levels $d'$ in parallel at time $t$, but to different units). We now must show identification for the remaining interventions $d > 0$ levels that take place in our dataset after time $t$.

Assume the induction hypothesis that $t_s$ is the $s$-th unique time an intervention level $d > 0$ is assigned to some unit in the dataset, and all intervention levels $d^\star$ that appeared up prior to $t_d$ have their

functions $\psi_l(d^\star, \cdot, \cdot)$ identified. We showed this is the case for $t_1 = t$, which is our base case. We assume this to be true for $t_s$ and we will show that this also holds for $t_{s+1}$.

Let $d$ be an intervention level assigned at $t_{s+1}$ and let $\mathcal{N}_d$ the corresponding subset of data assumed to exist by Assumption 3. For any $n \in \mathcal{N}_d$, let $s_-$ be the last time a non-zero intervention $d_-$ has been assigned to $n$ prior to $s + 1$. Also by Assumption 3, this must have taken place at least $k_{d_-}$ steps in the past. By the induction hypothesis, $\psi_l(k_{d_-}, \cdot, \cdot)$ is identifiable, and therefore so is the case for all $\psi_l(k_{d_\prec}, \cdot, \cdot)$ for intervention levels $d_\prec$ taking place for the first time prior to $t_s$. This also implies that $\eta_{nl}^{t_s}$ is identified, and the rest of the argument proceeds as in the base case. $\square$

**Theorem 2** *Assume we have N calibration samples,*

$$S_n^{t'} = |X_n^{t'} - f(D_n^{1:t_n}, X_n^{1:t_n}, Z_n)|, \quad t' = t_n + \Delta < t, \quad n = 1, \dots, N \tag{32}$$

*where $t_n$ is the time user $n$ experienced the last intervention before $t$. Assume there exists $\epsilon > 0$ such that, for all $n$,*

$$p_T(S_n^{t+\Delta}) = (1 - \epsilon)p_C(S_n^{t+\Delta}) + \epsilon p_\delta(S_n^{t+\Delta}), \tag{33}$$

*where $p_T$ and $p_C$ are the (unknown) densities of the test and calibration distributions, $p_\delta$ is a bounded arbitrary shift density, and $p_{min} = \min_{n=1,\dots,N} p_C(S_n^{t'}) > 0$. Then,*

$$\mathrm{Prob}\left(X_{N+1}^{t+\Delta} \in C\right) \geq 1 - \alpha - \frac{1}{p_{min}}\frac{\epsilon}{1 - \epsilon}. \tag{34}$$

**Proof.** Let $P_T$ and $P_C$ be the joint distributions of $(X_n^{t'}, D_n^{t'})$ for $t' > t$ and $t' \leq t$ and $w_n^{t'} = \frac{p_T(X_n^{t'}, D_n^{t'})}{p_C(X_n^{t'}, D_n^{t'})}$. For any $t' > t$, we can estimate $P_T$ by weighting the calibration samples with $w_n^{t'}$, $n = 1, \dots, N$, i.e.

$$P_T(X, D) \approx Z^{-1} \sum_{n=1}^{N} w_n^{t'} \mathbf{1}((X, D) = (X_n^{t'}, D_n^{t'})), \quad Z = \sum_{n=1}^{N} w_n^{t'}. \tag{35}$$

Conditional on the calibration samples, the $(1 - \alpha)$-th empirical quantile of $P_T(X, D)$ is

$$Q_T(1 - \alpha) = \inf_q \left\{ \left( Z^{-1} \sum_{n=1}^{N} w_n^{t'} \mathbf{1}(S_n \leq q) \right) \geq 1 - \alpha \right\} \tag{36}$$

$$= \inf_q \left\{ \left( \sum_{n=1}^{N} \left( \frac{w_n^{t'}}{Z} - \frac{1}{N} + \frac{1}{N} \right) \mathbf{1}(S_n \leq q) \right) \geq 1 - \alpha \right\} \tag{37}$$

$$= \inf_q \left\{ \left( \frac{1}{N} \sum_{n=1}^{N} \mathbf{1}(S_n \leq q) \right) \geq 1 - \alpha - \sum_{n=1}^{N} \left( \frac{w_n^{t'}}{Z} - \frac{1}{N} \right) \mathbf{1}(S_n \leq q) \right\} \tag{38}$$

$$\geq \inf_q \left\{ \left( \frac{1}{N} \sum_{n=1}^{N} \mathbf{1}(S_n \leq q) \right) \geq 1 - \alpha - \sqrt{N} \sqrt{\sum_{n=1}^{N} \left( \frac{w_n^{t'}}{Z} - \frac{1}{N} \right)^2} \right\} \tag{39}$$

$$= Q_C(1 - \alpha - \mathrm{gap}), \tag{40}$$

where $\mathrm{gap} = \frac{\sqrt{N}}{Z}\sqrt{\sum_{n=1}^{N}(w_n^{t'} - \mathbb{E}[w^{t'}])^2} = \frac{N}{Z}\mathbb{V}[w^{t'}] = \frac{\mathbb{V}[w^{t'}]}{\mathbb{E}[w^{t'}]}$, with $\mathbb{E}[w^{t'}]$ and $\mathbb{V}[w^{t'}]$ being the empirical mean and variance of $w^t$ estimated on the $N$ calibration samples. Under the theorem assumptions, we have $w_n^{t'} = (1 - \epsilon) + \epsilon\frac{\delta(S_n^{t'})}{p_C(S_n^{t'})}$. Then, $\mathbb{E}[w^{t'}] = 1 - \epsilon + \epsilon\frac{\mathbb{E}[\delta(S^{t'})]}{p_C(S^{t'})} \geq 1 - \epsilon$ and

$\mathbb{V}^2[w^{t'}] = \mathbb{E}[w^{t'} - \mathbb{E}[w^{t'}]]^2 \leq \epsilon^2\mathbb{V}^2\left(\frac{\delta(S^{t'})}{p_C(S^{t'})}\right) \leq \frac{\epsilon^2}{(\min_S p_C(S^{t'}))^2}\mathbb{V}^2[\delta(S^{t'})] \leq \frac{\epsilon^2}{(\min_S p_C(S^{t'}))^2}$.

The statement follows from

$$\mathrm{Prob}(Z \leq a) \geq \mathrm{Prob}(Z \leq b) \quad \text{if} \quad a \geq b, \tag{41}$$

and the Quantile Lemma on $S_{N+1}^{t+\Delta}$, which implies $\mathrm{Prob}(S_{N+1}^{t+\Delta} \leq Q_C(1 - \alpha_*)) \geq 1 - \alpha_*$, where $\alpha_*$ obeys

$$\alpha_* = \alpha + \mathrm{gap} \leq \alpha + \frac{\sqrt{\mathbb{V}^2[w^{t'}]}}{\mathbb{E}[w^{t'}]} \leq \frac{\epsilon}{\min_S p_C(S)}\frac{1}{1 - \epsilon}. \tag{42}$$

$\square$

## C  Simulators

We design two simulators: (1) a fully-synthetic simulator; and (2) a semi-synthetic simulator.

### C.1  General Setting

The simulators serve as fully controllable oracles to allow us to test the performance of our predictive causal inference problems. In particular, we have the following parameters:

- $N$: the total number of training users.
- $M$: the total number of testing users.
- $T$: the total number of steps in a time series.
- $T_0$: the number of steps in a time series when the intervention is the "default" one ($D = 0$).
- $K$: the number of different interventions (so that $D \in \{0, 1, \cdots, K-1\}$).
- $r$: the dimensionality of the feature space.
- $z_{\text{dim}}$: the number of time-invariant covariates.
- $I$: the maximum number of non-zero intervention levels in a time series.

Whenever possible, we set the same random seeds of $1, 2, 3, 4, 5$ to aid reproducibility of results. For the fully-synthetic simulator, a different seed indicates that it is a different simulator (as given by random draws of simulator parameters based on this seed). This also reflects the randomness coming from neural network parameter initialization, and data splitting when training models. For the semi-synthetic simulator, the simulator parameters $\phi$ and $\beta$ are learned from real world data, but we will need to also draw synthetic parameters for $\psi$. In both cases, using different random seeds can be considered as drawing new problems where each has unique parameters.

### C.2  Fully-Synthetic Simulator

The first simulator is purely synthetic, where all parameters are randomly sampled from some pre-defined distribution. The overall data generation process exactly follows the model we specified in Section 2.

We generate a synthetic dataset for training with the following parameters: $r = 5$, $T = 20$, $T_0 = 10$, $K = 5$, $N = 50,000$, and $z_{\text{dim}} = 5$. We generate $200,000$ samples at first place, but only use the first $50,000$ samples for training. To test our performance, we generate additional non-overlapping $M = 5,000$ samples with the same parameters, but this time with $T = 25$. We make predictions on the last $5$ time steps based on the first $20$.

The intervention effect parameters are sampled from the following distributions, each with a size of $K \times r$. For $d = 1, 2$ :

$$w_{1d} \sim \mathcal{U}(1, 2)$$
$$w_{2d} \sim \mathcal{U}(1, 2)$$
$$w_{3d} \sim \mathcal{U}(2, 3),$$

where $\mathcal{U}(a, b)$ is the uniform distribution in $[a, b]$. For $d = 3, 4$:

$$w_{1d} \sim \mathcal{U}(-2, -1)$$
$$w_{2d} \sim \mathcal{U}(-2, -1)$$
$$w_{3d} \sim \mathcal{U}(-1, 0).$$

We set $w_{1d} = 0$, $w_{2d} = 0$ and $w_{3d} = 1$ to reflect the "idle" or "default" intervention level $d = 0$. Values are designed to symmetric around the default intervention value. We randomly generate time-series $\{D_n^t\}$ based on the identification results in Section 2.2, with the following rules:

1. we have enough default-intervention points ($T_0 = 10 \geq r$, see Assumption 1).

2. for each unique unit, non-zero action levels are sampled without replacement. We additionally assume at least $I > K/2$ (in this case, $I = 3$) of the interventions show up in each unit-level time series.

3. once a $d > 0$ action happens, there must be at least two consecutive $d = 0$ actions (see Theorem 1).

4. in the test dataset, at prediction time, we assume an unseen intervention in the first 20 time series shows up at exactly step 21, and then the time-series has intervention level of 0 until the end. For instance, the training sequence is $D = [0, 0, \cdots, 1, 0, \cdots, 0, 2, 0]$ with $T = 20$, and test sequence is $D = [0, 0, \cdots, 4, 0, \cdots, 0, 3, 0] + [2, 0, 0, 0, 0]$ with $T = 25$.

We generate a time series $\{\psi_n^t\}$, based on Eq. (2), using the corresponding $w_{1d}, w_{2d}, w_{3d}$ values. We generate $r$ iid entries $\beta_{nl} \sim \mathcal{N}(\mu_n, \sigma_n)$ for each user, as follows:

$$\mu_n \sim \mathcal{N}(0, 1)$$

$$\sigma_n \sim \exp\{\mathcal{N}(0, 1)\}.$$

For each user $n$, we generate $z_{\text{dim}} = 5$ iid unit-specific covariates $\{Z_n\}$ as:

$$Z_n \sim \mathcal{N}(0, 3).$$

The same is used to set up an initial $X_n^0$ (the initial starting point of the time series) with a random-parameter MLP with input dimension of $z_{\text{dim}}$ and output dimension of 1. Finally, to create the synthetic time series $\{X_n^{0:T}\}$ (0 comes from the initial starting point), we randomly generate a neural network consisting of a single long-short term memory (LSTM) layer followed by a MLP to generate $\phi$ at each time step. We further compose $\phi$ with a sigmoid function so that it is bounded between $[-1, 1]$. We iteratively sample the $\{X_n^t\}$ points with the conditional mean from Eq 1 (based on $X_n^{0:t-1}$, $Z_n$, and $D_n^{1:t}$) plus a standard Gaussian additive error term at each step.

The outcome of this simulator gives us the following data: time-series $\{X_n^{0:T}\}$, intervention sequence $\{D_n^{1:T}\}$ and covariates $Z_n$.

## C.3 Semi-Synthetic Simulator

The second simulator is semi-synthetic, in the sense that the simulator parameters are learned based on real world data from the WSDM Cup, organised jointly by Spotify, WSDM, and CrowdAI[4]. The dataset comes from Spotify[5], which is an online music streaming platform with over 190 million active users interacting with a library of over 40 million tracks. The purpose of this challenge is to understand users' behaviours based on sequential interactions with the streamed content they are presented with, and how this contributes to skip track behaviour. The latter is considered as an important implicit feedback signal.

For a detailed description of the dataset, please refer to [10]. We created our simulator based on the "Test_Set.tar.gz (14G)" file. After unzipping the test data, we randomly chose a file named ("log_prehistory_20180918_000000000000.csv") as our main data source. In our case, we only use the columns named: "session_id", "session_position", "session_length", "track_id_clean", and "skip_3". Here, each unique "session_id" indicates a different sequence of interactions from an unknown user. The length of the interaction is denoted by "session_length", where each step is denoted by "session_position". The column "track_id_clean" indicates the particular track a user listens to at each position in the process. "skip_3" is a boolean value indicating whether most of the track was played by the user. We assign the number of 1 if it is "True" and 0 otherwise. We use this value to later create $X_n^t$.

To process the data, we start by looking at the existing sequence lengths, and select the most frequent sequence length in the data (20) (the actual corresponding session sequence is 10). This accounts for approximately $58\%$ of the original file. After filtering, there are around $228,460$ unique sessions and $301,783$ unique tracks in the data. To further reduce the computational cost of constructing a very large session-track matrix, we randomly sample a sub-population of $6,000$ unique session ids, which then bring us to $26,707$ unique tracks. We build the session-track matrix where the entry value is the number of counts a particular track has been listened to over a particular session based on this sub-population. We apply singular-value decomposition (SVD)[6] to this session-track matrix to get session embeddings and track embeddings. For numerical stability reasons, we save

---

[4]https://research.atspotify.com/datasets/

[5]https://open.spotify.com/

[6]https://numpy.org/doc/stable/reference/generated/numpy.linalg.svd.html

only the top 50 singular values and further normalize them using its empirical mean and standard deviation. For track embeddings, we take the top 10 singular values after the normalization and assign them into 10 clusters, using constrained $K$-means to to encourage equal-sized clusters[7]. Using these clusters, we create a new categorical variable $F$ and its continuous counterparth $\phi(F)$, which respectively represent the corresponding cluster and its centroid vector based on the value of its associated "track_id_clean" entry. Next, we set $X_n^t$ as the cumulative sum of the corresponding binary "skip_3" value over each unique "session_id" with added zero-mean Gaussian noise with standard deviation $1/2$. We keep the first 5 singular values for the session embedding normalization and consider them to be the session specific covariate $Z$. We save $X$, $F$, $\phi(F)$ and $Z$ as separate matrices as the final output, which have shapes of $6,000 \times 10$, $6,000 \times 10 \times 10$, $6,000 \times 10 \times 10$ and $6,000 \times 5$, respectively.

To build the simulator, we further define the following two processes:

$$P(F_n^t \mid F_n^{1:t-1}, X_n^{1:t-1}, Z_n) = \text{softmax}(f(F_n^{1:t-1}, X_n^{1:t-1}, Z_n)),$$

and

$$X_n^t = \phi(F_n^t)^\mathsf{T} \beta_n + \epsilon_n^t,$$

where $\phi(F_n^t)$ is the 10-dimensional embedding of categorical variable $F_n$ given by the cluster centroid, and $\epsilon_n^t \sim N(0, \sigma_x^2)$.

Function $f(\cdot, \cdot, \cdot)$ is modeled with a deep neural network consisting of a single layer GRU and a MLP, trained on the real $F$ with cross entropy loss (as a likelihood based model). The second equation is trained on the real $X$ with an ordinary least square (OLS) regression with a ridge regularization term ($\alpha = 0.01$). Error variance is estimated by setting $\hat{\sigma}_x^2$ to be equal to the variance of the residuals. This defines a semi-synthetic model calibrated by our pre-processing of the real data.

We now need to define interventions synthetically, as they are not present on the real data. We choose $K = 5$ as the size of the space of possible interventions. We will define interventions in a way to capture the notion of changing the exposure of tracks in particular cluster $k$ for particular user $n$ at time $t$. Since the real data does not contain information regarding how interventions can influence the behaviour of Spotify users, similarly to the simulator setup for the fully-synthetic one, we use 5 random seed to indicate different possible changes to a user behaviour in the Spotify platform.

The intervention effect parameters are sampled from the following distributions, each with a size of $K \times r$. For $d = 1, 2$ :

$$w_{1d} \sim \mathcal{U}(1, 1.5)$$
$$w_{2d} \sim \mathcal{U}(1, 1.5)$$
$$w_{3d} \sim \mathcal{U}(1.5, 2.0).$$

For $d = 3, 4$:

$$w_{1d} \sim \mathcal{U}(-1.5, -1)$$
$$w_{2d} \sim \mathcal{U}(-1.5, -1)$$
$$w_{3d} \sim \mathcal{U}(0, 0.5).$$

We set $w_{1d} = 0$, $w_{2d} = 0$ and $w_{3d} = 1$ to reflect the "idle" or "default" intervention level $d = 0$. The values are designed to symmetric around the default intervention value. We randomly generate time-series $\{D_n^t\}$ based on the identification results in Section 2.2 and the rules we adopted in the full-synthetic simulator in Appendix C.2. Notice that the above does not correspond exactly to the model class we discuss in Section 2, as the product of intervention features $\psi$ in Eq. (2) is pipeline with the softmax operation of $f(F_n^{1:t-1}, X_n^{1:t-1}, Z_n)$ to get $F_n^t$, which is then composed with $\phi$ to get $\phi(F_n^t)$.

We generate a synthetic training dataset with the following parameters: $r = 10$, $T = 35$, $T_0 = 25$, $N = 3,000$, and $z_{\text{dim}} = 5$ (notice that we have the first $5,000$ for training, but only use the first $3,000$ samples for the actual baseline training, as we have additional experiments to check the influence of more training data points). To test our performance, we generate additional non-overlapping $M = 1,000$ samples with the same parameters, but this time with $T = 40$ for making predictions on the last 5 time-series steps based on the first 35.

---

[7]https://github.com/joshlk/k-means-constrained

To sample data from this simulator, we take the initial state of $X_n^0$ and $F_n^0$ from the pre-processed data. We use the $Z_n$ vector which comes from the real-world user embedding. Then, we iteratively generate the next $F_n$ via categorical draws from the softmax output based on user embedding $Z_n$ all previous $F_n$ and $X_n$. Once a categorical value of $F_n$ is drawn, we replace it with its embedding form $\phi(F_n)$, which is the centroid of its cluster. We use this $\phi(F_n)$ to generate $X_n$ plus an additive Gaussian noise with zero mean and homoscedastic variance learned from data.

The outcome of this simulator gives us the following data: time-series $\{X_n^{0:T}\}$, intervention sequence $\{D_n^{1:T}\}$ and covariates $Z_n$.

# D    More Results

In this Section, we present more experimental results for our method. This includes: (1) the effect of choosing $r$; (2) further examples of visualization and empirical results.

## D.1    Effect of the Choice of $r$

The effect of different choices of $r$ is presented in Fig. 3, with $r = 5$ being the ground truth parameter from which the data is generated. We have also plotted the performance of our method under $r = 3$ and $r = 10$. We observed that when $r$ is smaller than the ground truth, the model tend to under-fit the data with a slightly higher MSE error. When $r$ is bigger than the actual value, the MSE error can be further reduced, but resulting in very large variance. We also show that this can be mitigated with further regularization, such as applying $L_1$ and $L_2$ penalty terms during the likelihood learning process.

The insight we draw from this is that, when applying our approach to real-world problems, we could potentially choose a higher $r$ without much concern about what an exact value should be. Using a large $r$ leads to higher variance but in general performs reasonably well compare to knowing the true $r$ (here, $r = 5$). We also notice that applying regularization techniques, even when we know the right dimension of $r$, can be beneficial. This leads to a more stable performance (that is, with lower variance), but does not change much of the mean estimate as shown in Fig. 3. This maybe partially caused by the gradient optimization process when training the deep neural network. Hence, we claim that the main price to be paid for a large choice of $r$ is that this makes data requirements stricter (such as the number of steps prior to the first intervention) in order to guarantee identification.

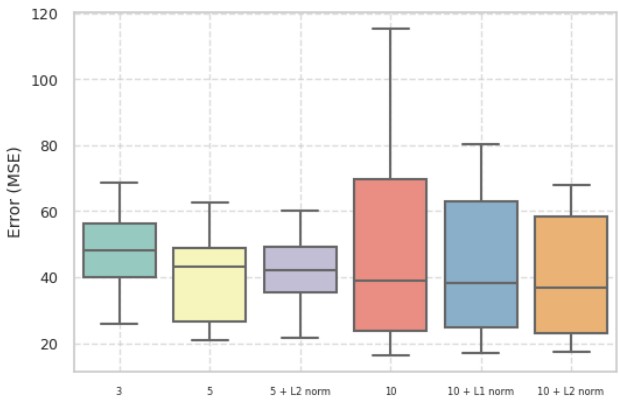

Figure 3: Effect of changing $r$ for the fully-synthetic dataset.

## D.2 Additional Figures

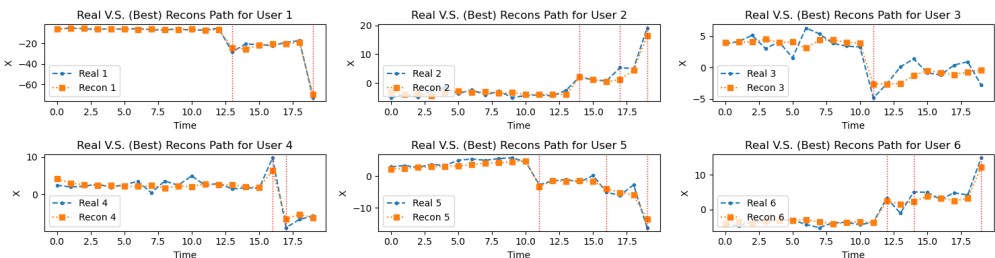

Figure 4: Examples of reconstruction of training data for CSI-VAE-1 model.

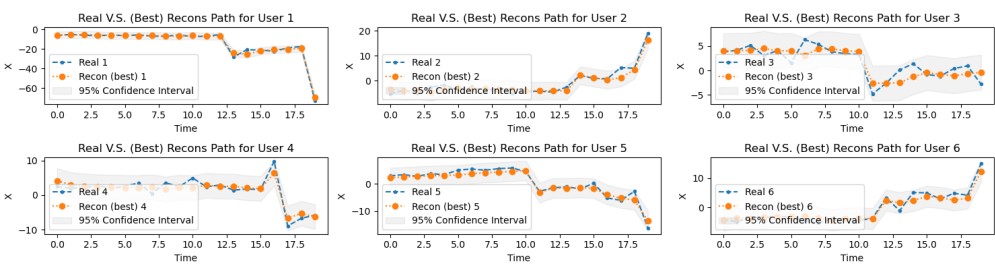

Figure 5: Model-based uncertainty quantification of reconstruction for CSI-VAE-1.

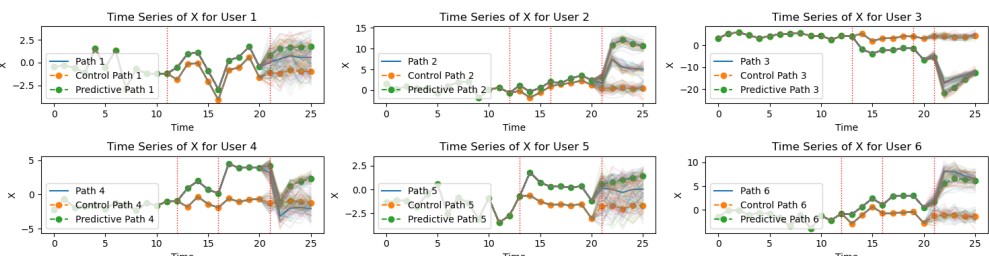

Figure 6: Demonstration of prediction, in the synthetic data case, for CSI-VAE-1.

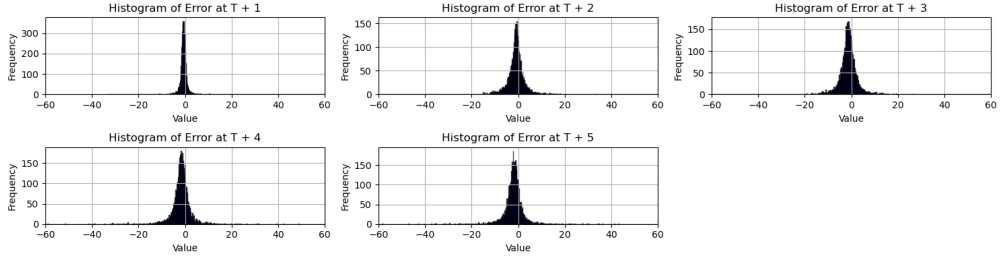

Figure 7: Residual distribution examples for CSI-VAE-1. In general, we observe a very long tail effect across model predictions.

## E  Uncertainty Quantification

For the sake of illustration, we generate another new dataset with a random seed of $42$ using the synthetic simulator, based on the following specifications: $r = 5$, $T = 20$, $T_0 = 10$, $K = 5$,

$N = 50,000$, $z_{\dim} = 5$, $I = 3$. We also generated two further datasets of $M = 2,000$, one for calibration and one for testing.

## E.1 Plug-in Model-based Uncertainty Quantification

We first present the model-based uncertainty quantification at level of $\alpha = 0.95$, in Fig. 8. Given estimates of the conditional mean and homoscedastic error variance $\sigma^2$ learned from data, we can calculate predictive intervals as $(\mu - 1.96 \times \sigma, \mu + 1.96 \times \sigma)$. For purely model based uncertainty quantification with plug-in parameter estimates, we do not need an additional calibration dataset, but it is an obviously problematic one as it does not take into account estimation error and sampling variability. The coverage rate is $57.75\%$ on the test dataset without further calibration.

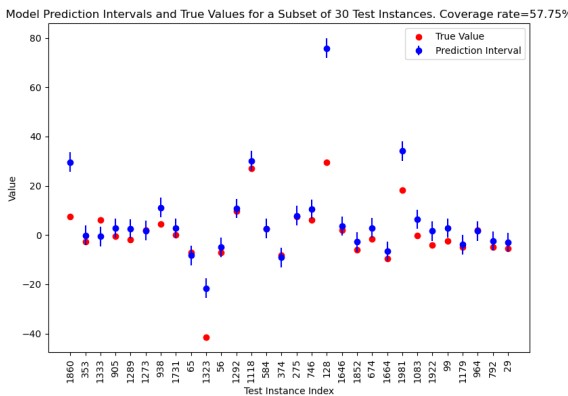

Figure 8: Plug-in model-based predictive interval and coverage.

## E.2 Conformal Prediction

We then present the model-free uncertainty quantification at level of $\alpha = 0.95$ (based on the Setup 1 in Section 3.2), shown in Fig. E.2. We use the absolute value of the predictive residual as our conformity score function, $|y - f(x)|$. We calibrate our model output based on the calibration dataset and then apply it on the test dataset. We obtained a coverage rate of $95.30\%$. This is a significant improvement upon what we see in Fig. 8 and demonstrates the effectiveness of the conformal prediction approach.

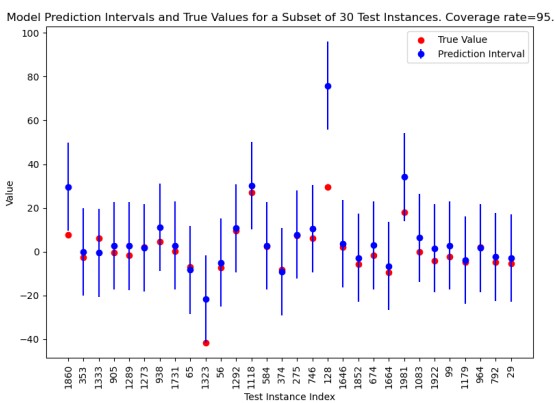

Figure 9: Conformal prediction predictive interval and coverage.

