# OpenReview forum: "Structured Learning of Compositional Sequential Interventions"
_NeurIPS.cc/2024/Conference — NeurIPS 2024 poster_

### Official Review · Reviewer_MJr7 · 2024-07-10

**Soundness:** 3
**Presentation:** 2
**Contribution:** 3
**Rating:** 7
**Confidence:** 2

**Summary:**

Estimating the causal effect of a sequence of interventions on another sequence is a central task in causal inference / treatment effect estimation. However, for large discrete spaces, canonical assumptions such as Markovian assumptions, short sequences, and so forth will not apply, while very general black-box models may present poor generalizability due to the sparsity of the observed treatment sequences. This work presents a model where the effect of a sequence of interventions on the target sequence can be decomposed as a product of individual time-point-wise effects. This parameterization is fairly general as it parameterizes functions of a specific form exactly, and any measurable function approximately. Identification results are given for the parameter of the proposed parameterization. This motivates a VAE-based fitting approach, together with predictive intervals coming from conformal prediction. Experiments are given, showing competitive performance of the method.

**Strengths:**

The work is original as it shows that coming with a specialized parameterization can help, both theoretically (identifiabiity) and empirically (better performance), than a general black-box model. It is also significant as the parameterization is actually not particularly restrictive, as Propositions 1 and 2 show that it can recover exactly or approximately general types of functions. Thus the method is generally very principled and should be very applicable in the field.

**Weaknesses:**

1) IMO the main drawback of the work is in clarity.

1a) It took me significant time to parse what the field is missing and what is the exact contribution of the method. The contribution is specified in the "Scope" paragraph l.37-42 and later in l.64-78, and the previous work with its drawbacks is scattered in l.43-63, Section 4 l.304-322 and Appendix A. I think that these sections should be completely re-organized, notably all the previous work should be put together and the contributions outlined clearly in contrast to such previous work ; especially as the contribution consists actually in specifying a more restricted functional form that's more tailored to a specific type of intervention sequence.

1b) It is not clear what the exact estimand is, or what the exact estimands are, as they are not stated in the problem statement l.79-90. The functional form $\mathbb{E}[X_n^t(d_n^{1:t}) | x_n^{1:t-1}(d_n^{1:t-1}), z_n]$ is also confusing, as it suggests that it does not depend on covariate values but only on $n$, $t$ and interventions, while from l.173 it also depends on covariates. It is only in Section 4 that I understood that I understood that only predictions were to be made conditional on previous covariates. Thus, I would be happy if authors could state in their rebuttal what estimand(s) are we looking for, and if they could write it here.

1c) It is also not clear how the functional form of Equation 1 incorporates "compositionality" of interventions. While it can be guessed from Equation 2, it is only in l.184-192 that it becomes clear this functional form for $\psi$ incorporates compositionality (if I understood it well?)

2) While experiments generally suggest the relevance of the method, I'd have a few concerns :

2a) There seems to be rather few baselines : GRU-0 and GRU-1 seem rather redundant as they look like mutilated versions of the GRU-2 baseline ; the latter seems like the only actually fair comparison against CSI-VAE as only that baseline takes the whole intervention history into account. The bottom part of Figure 2 indirectly confirms this as only GRU-2 is evaluated. Thus, all of this might leave GRU-2 as the only real baseline. Could another one be implemented?

2b) Conformal prediction is incorporated in the paper, but it is only evaluated in the Appendix and only on the method. Could an analysis include baselines and be moved to the main part?

Note : I am aware that it might not easy to obtain new empirical results in the rebuttal period, I am happy to discuss all of this with other reviewers and ACs

**Questions:**

1) I struggle to understand what these mean, can you develop a bit?

a) l.38-39 : "including the bursts of non-stationarity that the application of an intervention brings in"

b) l.106-108 : "We do not explicitly condition on $D_{1:t}^n$ in most of our notation, adopting a convention where potential outcome indices $D_{1:t}^n$ are always in agreement with the (implicit) corresponding observed $D_{1:t}^n$".

2) l.173 : how exactly does $\mathbb{E}[X_n^t(d_n^{1:t}) | x_n^{1:t-1}(d_n^{1:t-1}), z_n]$ depend on $x^{1:T}$ exactly? Which components of the latter are used?

3) How stringent are Assumptions 3 and 4?

4) Section 3.1 :
a) Is identifiability preserved with variational inference in CSI-VAE?
b) What is $\mu$ in the equation of l.250, and how does this incorporate the functional form of Equation 1?

**Limitations:**

Limitations have a dedicated paragraph, and social impact is no different than any other work on treatment effect estimation more broadly.

**EDIT (2024/08/11)** : increased my score following authors' rebuttal and discussion.

---

> ### Author Rebuttal · Authors · 2024-07-31
>
> Thank you for your review and your remarks that **“The work is original…”** and that is its **“generally very principled and should be very applicable in the field”**! The many comments about presentation are very useful.
>
> **IMO the main drawback of the work is in clarity. 1a.: …time to parse…1b. It is not clear what the exact estimand is…**
>
> The goal is "behavioral forecasting under hypothetical future interventions" (lines 41-42), where "behaviour" is defined in line 25.  We want (line 87) to predict $X_{n^\star}^{T + 1:T + \Delta}(d_{n^\star}^{1:T + \Delta})$ using whatever is observed up to time $T$, extending the past $D_{n^\star}^{1:T}$ with a hypothetical plan of future actions $D_{n^\star}^{T + 1:T + \Delta}$.
>
> We recognize that, even though this description fits standard machine learning formulations, it is typical for a causal modeling paper to have equations of estimands presented explicitly, even if this is not natural when the problem is predictive. To address this, we propose bringing predictive potential outcomes in terms of functionals earlier on in the write-up.  Under the usual mean-squared error loss, this commonly boils down to the expected value of each $X_{n^\star}^{T + 1}$, $X_{n^\star}^{T + 2}$, etc. given the past up to $T$. We originally didn’t want to prescribe this because one may want to use a different loss function other than mean squared error, where plain expectations wouldn’t be the answer. So, it’s not that straightforward to introduce a single equation such as e.g. $\mathbb E[X_n^{T’}(d_n^{1:T’}) |  x_n^{1:T}(d_n^{1:T}), z_n]$ for $T’ > T$ as the target estimand, because *we are not particularly committed to any given functional of the predictive distribution*: we only want to deal with (potential) observables, not with functionals (that’s why we focus on conformal prediction instead of confidence intervals, and predictive potential outcomes as opposed to cross-world causal effects).
>
> Nevertheless, we should anticipate that some readers (like the reviewer) may find it jarring not to have an explicit estimand equation laid out right at the beginning. Given that Eq. 1 is presented just for the predictive mean, we can use this as the family of estimands. So in a revised manuscript we will further emphasize the predictive aspect within the context of means (motivated by mean squared losses) earlier on.
>
> (we have one question for the reviewer: we didn’t quite get what the reviewer meant by “... it suggests that it does not depend on covariate values but only on $n$, $t$,  and interventions, …” - the functional form listed seems to be the same as in l.173?)
>
> **1c) It is also not clear how the functional form of Equation 1 incorporates "compositionality" of interventions...  (if I understood it well?)**
>
> The reviewer got it correctly, but as mentioned it won’t hurt to anticipate early on in the text where we are going with compositionality, including the function composition of Eq. 4.
>
> **Experiments**
>
> We focused on the GRU because it is a well established method for learning sequential predictions out of categorical input sequences - more flexible than RNNs, less complex than LSTMs or transformers, which would be major overkills: we already overfit with GRUs, a LSTM or beyond would not help (and did not, in preliminary experiments). We nevertheless provide updated experiments in the shared rebuttal box.
>
> GRU-0 and GRU-1 should be seen as ablation studies. GRU-0 quantifies the total contribution of the signal coming from $D$, showing that results are not just artifacts of modeling the evaluation of the $X$ series. GRU-1 excludes long-term histories, focusing only on the latest iteration of intervention, and illustrates that a very strong Markovian structure cannot emulate long-term direct effects. Doing this in the context of GRU shows that even a flexible model that potentially overfits nevertheless benefits from long-term, non-Markovian contributions of past interventions.
>
> Regarding conformal prediction, as the results are done in (semi)synthetic experiments we expect theory to agree with, and so they are in less need to be thoroughly empirically assessed. But we will do our best to include them in the main text in the eventual acceptance of the paper, which would allow us to have an extra page.
>
> **1a) l.38-39**
>
> When an intervention is applied, we expect a system to behave in a non-stationary way in the short-term, eventually settling down to a possibly new equilibrium. See for instance Fig. 1 of [9].
>
> **1b) l.106-108**
>
> We do not allow conditioning of $X^{1:t}$ generated under different $D^{1:t}$ other than the one used in forecasting future trajectories i.e. no cross-world counterfactuals.
>
> **2. l. 173**
>
> They are used in the definition of $\phi_{nl}^t$ (see line 109).
>
> **3. Stringency Assumptions 3 and 4.**
>
> Assumption 4 is relatively benign, basically that we don’t have linear dependency between (features of) past trajectories.
>
> Assumption 3 essentially requires that there are enough units of time for a large enough group of individuals to be exposed to a particular treatment level $d$ prior to being exposed to something else. It will depend on how smooth a treatment effect is. Citing again Fig. 1 of [9], realistic effect shapes in many domains can be described with relatively few parameters (effect smoothly going up, then down, then settling), which should be amenable to this assumption holding in practice.
>
> **4. Section 3.**
>
> The VAE still uses a likelihood (decoder) where the conditional mean is given by Eq. 1, and the conditional variance is homoscedastic and easily shown to be identifiable. Identifiability of the parameters of the encoder is not fundamental (it just represents a posterior distribution, it does not specify a causal structure) and its variance will go to zero as the number of time points increases.
>
> Many thanks again for the useful questions!

---

> > ### Comment · Reviewer_MJr7 · 2024-08-09
> > **Answer to Aug 7 rebuttal**
> >
> > Many thanks to authors for the extensive rebuttal. My questions have been answered, except the following points :
> >
> > 1) I still do not understand exactly how $E[X_n^t(d_n^{1:t})|x_n^{1:t−1}(d_n^{1:t−1}),z_n]$ in Equation 1 and $\phi_l(x_n^{1:t−1}(d_n^{1:t−1}), z_n)$ in Equation 109 depend on $x_n^{1:t−1}, d_n^{1:t−1}, z_n$, or more precisely what the notation $x_n^{1:t−1}(d_n^{1:t−1})$ means. It suggests that $x_n^{1:t−1}$ is a function of $d_n^{1:t−1}$, while of course also depending on $n$ and $t$. Further, from this interpretation, it is at first unclear whether this function is known a priori (before observing the data) or pre-specified, or it is observed in the data, or it has to be learnt, etc... This is what I mean in the statement raised by your own question "we have one question for the reviewer: we didn’t quite get what the reviewer meant by" : if $x_n^{1:t−1}$ is a function that is pre-specified or known a priori, then indeed $x_n^{1:t−1}(d_n^{1:t−1})$ only depends on $d_n^{1:t−1}$ in the argument, and $n,t$ in the index.
> >
> > It is later in the paper that I understood that these $x_n^{1:t−1}$ are actually values taken by the (potential) covariates, thus these $x_n^{1:t−1}$ are vectors or scalars but not functions (if I am not mistaken). Thus, it now seems to me that the $x_n^{1:t−1}(d_n^{1:t−1})$ notation for small $x$ is incorrect and should be scraped altogether and replaced with either $X_n^{1:t−1}(d_n^{1:t−1})$ with capita $X$ in statistical quantities, and $x_n^{1:t−1}, d_n^{1:t−1}$ in functions. To be more precise, I understand that one should write
> >
> > a) $E[X_n^t(d_n^{1:t})|X_n^{1:t−1}(d_n^{1:t−1}) = x_n^{1:t−1}, z_n]$ in Equation 1, which makes it clear that $X_n^{1:t−1}(d_n^{1:t−1})$ is a potential covariate that is indexed at the observed intervention $d_n^{1:t−1}$ and is here taking the observed value $x_n^{1:t−1}$ ;
> >
> > b)  $\phi_l(x_n^{1:t−1}, d_n^{1:t−1}, z_n)$ in Equation 109, as this is a function as in l.173 and it does not formally rely on random variables, including potential covariates.
> >
> > Is all of this correct? If not, is $x_n^{1:t−1}$ for a small $x$ actually a function of $d_n^{1:t−1}$?
> >
> > 2) My mistake, I had misread Section 3.1 and missed the point that the mean and conditional variance of l.250-251 are actually the posterior parameters of $\beta_n$ (feel free to correct me if I am wrong here). Here are some questions, mostly for clarity:
> > a) Are these parameters for the encoder then?
> > b) By the conditional mean and variance of the decoder, do you refer to Equation 233?
> > c) Do you also confirm that the VAE departs from the setup of Theorem 1, as notably the basis function is learnt?

---

> > > ### Author Response · Authors · 2024-08-10
> > > **Thank you**
> > >
> > > Thank you, this follow-up is super-useful to understand where you (and other potential readers) are coming from! Yes, we used $x_n^{1:t - 1}(d_n^{1:t - 1})$ as notation for a realization of $X_n^{1:t - 1}(d_n^{1:t - 1})$. Although not unsound, we can see how this may invite unnecessary confusion. Even though using $v$ as short-hand notation for the event $V = v$ when following a conditioning bar is standard, the extra piece of notation about the potential outcome index gets in the way and it's not very common. We wouldn't want to use $\mathbb E[X_n^t(d_n^{1:t}) | X_n^{1:t-1}(d_n^{1:t-1}), Z_n]$ (capital letters) as later on in e.g. Proposition 3 we mention realizations, but  $\mathbb E[X_n^t(d_n^{1:t}) | X_n^{1:t-1}(d_n^{1:t-1}) = x_n^{1:t-1}, Z_n = z_n]$ will help to establish a convention about the implicit regime invoked when referring to $x_n^{1:t-1}$, including our references for $\phi_l$ (where we do need to instantiate as $\phi_l(x_n^{1:t-1}, d_n^{1:t-1}, z_n)$ when defining $\phi_{nl}^t$ in line 109). Thank you!
> > >
> > > Concerning your point 2, yes those are posterior parameters. They are not independent parameters of the encoder per se, strictly speaking the parameters are the ones in the MLP/GRU pipeline (lines 250-251), which then define the mean and variance of the variational Gaussian distribution in the usual amortized variational inference sense. Line 233 is the likelihood function indeed, where $f(\cdot)$ is the conditional mean as in Eq. 1 and the error variance is just another parameter we can optimize with respect to. Finally, regarding your point c), our theory can be respected by basically freezing $\phi$ after the initial $T_0$ period (in this case, $\beta$ is considered identified with respect to the learnt $\phi$ -- it's not fundamental that a different $\phi$ would imply a different $\beta$, the choice of basis $\phi$ is problem-dependent anyway). In practice though, as we say in line 258, we allow our implementation to just backprop through $\phi$ even after $T_0$ (based on preliminary experiments with different simulations), although this can be easily switched off.
> > >
> > > Many thanks again, very useful to get suggestions on how to tweak the notation to increase accessibility. Good stuff!

---

> > > > ### Comment · Reviewer_MJr7 · 2024-08-10
> > > > **Answer to Aug 9 answer**
> > > >
> > > > Many thanks for your new answer. All is clear now. I will increase my score and more firmly recommend admission.

---

> > > > > ### Author Response · Authors · 2024-08-11
> > > > > **Thank you**
> > > > >
> > > > > Thank you very much for engaging with us!

---

### Official Review · Reviewer_2QJb · 2024-07-10

**Soundness:** 3
**Presentation:** 2
**Contribution:** 2
**Rating:** 5
**Confidence:** 3

**Summary:**

This paper considers a special case of a series of interventions where the interventions are categorical and sparse. And interventions can effect later timestamp. This is a form of causal extrapolation. Authors propose to study this using a conditional mean model that utilizes basis functions and subsequently develop corresponding algorithms.

**Strengths:**

1. The problem is interesting and well-motivated.
2. As far as I can tell, the theory is sound.

**Weaknesses:**

1. Section 2.1 is really hard to parse.
2. The assumptions in section 2.2 is not explained intuitively and I don’t know how necessary they are.
3. The experiments are synthetic and semi-synthetic. Although since the nature of the paper is mostly theoretically, I wouldn’t consider that to be a huge issue.

**Questions:**

1. In equation (1), do you need the lower case x to be a potential outcome?
2. Is the notation changed in 2.1? f does not appear in equation (1).
3. For CSI-VAE, why use GRU instead of more modern arch like attention or even LSTM?

**Limitations:**

The only limitation is how practical these methods are.

---

> ### Author Rebuttal · Authors · 2024-07-31
>
> Thanks for your review, and for agreeing that **“the problem is interesting and well-motivated”**! We would like to see more of that in the community.
>
> **Section 2.1 is really hard to parse**
>
> We hope the following helps. Eq. 1 takes a standard tensor decomposition format. It can be motivated by a variety of starting points, for instance Taylor series approximations. Its main hyperparameter is $r$, for which there are standard ways of choosing it - see for instance [1]-[4].
>
> In Section 2.1, we address it from two perspectives. *First*, when time is unbounded: there we assume that the true model can be constructed from the exact form in line 149, as motivated by results such as Proposition 1 of [25], which can then be transformed into the structure of Eq. 1. The reasoning is formalized as Proposition 1 in our paper.  *Second*, when time is bounded: we can allow the true data generating functions to belong to a broad class of functions, showing that there exists a finite $r$ that controls the approximation error of Eq. 1 to any a priori degree. This is formalized as Proposition 2 in our paper.
>
> In particular, separating treatment variables from the rest, as in the equation in line 135, can be motivated as follows. First, by recognizing that this is a standard way of writing a regression function with categorical inputs. For instance, if we have two categorical inputs $d_1 \in \{0, 1, 2\}, d_2 \in \{0, 1\}$, we can have the following (overcomplete) ANOVA model with parameters $\theta_{00}, \theta_{01}, \theta_{10}, \theta_{11}, \theta_{20}, \theta_{21}$, with indicator functions $f_{(d_1’, d_2’)}(d_1, d_2) \equiv I(d_1 = d_1’, d_2 = d_2’)$:
>
> $$f(d_1, d_2) \equiv f_{(0, 0)}(d_1, d_2) \times \theta_{00} + \dots + f_{(2, 1)}(d_1, d_2) \times \theta_{21}$$
>
> When there are other inputs beyond the categorical variables of interest, we can make each $\theta$ a function of those.
>
> Similar constructions appear in many places, e.g. energy functions, exponential families, but also all sorts of machine learning methods - as we don’t want to span all exponentially many combinations of the discrete space. In a regression tree with categorical inputs, for instance, we don’t use all combinations, but only those that correspond to paths from the root to a leaf in some tree, with $\theta$ being the piecewise constant expected value of the output at inputs mapped to that leaf. This is a standard interpretation that appears in e.g. Hastie et al.’s *Elements of Statistical Learning*. One interpretation of what we do in Eq.1 is to use a differentiable embedding of the $d$ vector with dimension $r$, instead of $r$ combinatorial paths down a tree of $r$ leaves.
>
> We are very happy to take follow up clarification questions on the above.
>
> **The assumptions in section 2.2 is not explained intuitively and I don’t know how necessary they are.**
>
> Assumption 1 allows a large enough window of time so that we can identify $\beta$ first by least squares. For least-squares to be well-posed we need that the corresponding rows in the observation matrix are linearly independent. This is what Assumption 2 states (pseudo-inverses basically mean the matrix that provides the least-squares projection).
>
> Assumption 3 states that, when we need to learn about intervention level $d$, then we need a sufficient number of units getting that assigned, and that we leave these units unperturbed by enough time so that we don’t have further interventions getting conflated with $d$. Here, “enough time” is formalized in terms of the number of intervention parameters $k_d$.
> Assumption 4 is analogous to Assumption 2, now applied to the identification of $\psi$: having identified $\beta$ and fixing $\phi$, this will become another least-squares problem where again we need an explicit assumption about the rank of the matrix of observations.
>
> Let us know whether the above is helpful.
>
> **The experiments are synthetic and semi-synthetic. Although since the nature of the paper is mostly theoretically, I wouldn’t consider that to be a huge issue.**
>
> That was our reasoning too. We believe that the main point is to call the attention that off-the-shelf black-boxes may not be the wisest option here (a point we want more people to appreciate), so keeping the benchmark to be as uncluttered and controllable as possible was one of our main objectives. We nevertheless provide updated experiments in the shared rebuttal box.
>
> **In equation (1), do you need the lower case x to be a potential outcome?**
>
> At least implicitly it is necessary, since past $X$s must be exposed to the levels of past treatments as the future $X$s. We are not against other pieces of notation, such as having a single $do(d_n^{1:t})$ operator to indicate a single-world set of interventions. Sometimes it is slightly more convenient to have the potential outcome notation, as in the text of Assumption 4 where we refer to $\phi$ as explicitly generated at the specific intervention level $d_n^{1:t + t’ - 1}$, but it has some disadvantages too (such as a heavy syntax).
>
> **Is the notation changed in 2.1? f does not appear in equation (1).**
>
> It is changed to emphasize that in 2.1 we are talking about purely generic function representations - but yes, it is motivated to eventually have the shape of Eq. 1. In an eventual camera-ready version, we will modify the start Section 2.1 to provide a short summary of what will follow to facilitate reading.
>
> **For CSI-VAE, why use GRU instead of more modern arch like attention or even LSTM?**
>
> GRU was actually introduced several years after the LSTM as a way of simplifying it for applications where a full-blown LSTM was an overkill. We considered GRU to be better suited to our benchmarking, as we were not dealing with very long sequences and it was already overfitting - so little to be gained (and, actually, opening up the possibility of doing worse) by falling back into a LSTM, not to mention transformers.
>
> Thank you again for your time and feedback!

---

> > ### Comment · Reviewer_2QJb · 2024-08-12
> >
> > Thanks for you reply! I don't have any further question at this point.

---

> > > ### Author Response · Authors · 2024-08-13
> > > **Closing comments**
> > >
> > > Thanks again for your time and engagement. We hope we have addressed all questions about clarify and experiments.

---

### Official Review · Reviewer_yF1i · 2024-07-11

**Soundness:** 2
**Presentation:** 1
**Contribution:** 2
**Rating:** 3
**Confidence:** 4

**Summary:**

This paper studies treatment effect estimation under sequential, discrete interventions.
It is assumed that all treatments are fully independent and impact all future outcomes.
It studies the problem of "causal extrapolation" where some combinations of sequential interventions may not have been observed during training.
Parametric assumptions on the causal mechanisms are made and used for generalising to unseen interventions.
A matrix factorisation approach is developed to fill in the gaps for the unseen interventions.
The approach is empirically tested on synthetic and semi-synthetic data.

**Strengths:**

- This paper studies an important and, in my opinion, an understudied problem in causality: causal extrapolation, as it is called in the paper. Many papers look at assumptions that can be made on the causal structure alone, whereas here, the functional form of the structural equations is considered as well.
- The paper studies a way to generalize over unseen combinations of interventions. That is an important concern, especially in the time series context.

**Weaknesses:**

- The mathematical formalisation is very difficult to follow. Wild statements are made without properly introducing the objects, motivating the statement or clarifying what assumptions are made in order for the statement to hold (see questions below). It is unclear which parts follow from the assumption of the causal relationships in Fig. 1 and what additional assumptions or approximations are made.
- The assumptions on the data generating process are too restrictive for the method to have much hope of being applied to real data (see limitations).

**Minor:**

- L21 and L 86: Typo "an unit"

**Questions:**

- Eq. 1: Where does this come from? Can any treatment effect that follows the causal structure in Fig. 1 be expressed like this? Or is this a simplifying modelling assumption?
- L136: What are ANOVA models?
- L123: What does the last sentence in this paragraph mean?
- Equation in L135: What does this function correspond to? The equation seems not to hold for general functions: e.g. if the dimension of the two RHS functions is 1, then this is certainly not a statement that is true in general. Under what conditions can you decompose a function like this? What are the terms on the RHS? How do they fit into the initial problem setting?
- Prop. 3: Does this assume that equation 1 holds?

**Limitations:**

- Fig. 1: The assumed causal structure severely limits the applicability of the presented approach. There is no allowed confounding and, more importantly, the interventions are assumed to be independent. The latter assumption would require full randomisation during data generation. It does simplify things a lot, since it avoids having open paths through conditioning on colliders. But, in essence, it is assumed that the data comes from an RCT. Therefore, the contribution is applicable to RCTs that do not cover the full range of combinatorial interventions.
- The identifiability Assumption 1 means that we essentially have full access to probe each unit (we need to have the "no intervention" applied for several time steps). That is, we essentially can perform an RCT on all units. In that case, why not keep the units in the lab, rather than predicting what will happen under future interventions?

---

> ### Author Rebuttal · Authors · 2024-07-31
>
> Thanks for your review, and for the much appreciated point that the problem is **“important and, … an understudied problem in causality”**, which is one of the main messages we wanted to convey! We address all clarification questions below.
>
> **Eq. 1: Where does this come from?**
>
> This is an excellent point, which is why we dedicated the entire Section 2.1 of our paper solely about the representation power of Eq. 1. For a detailed answer to your question, please consult it. Here is a summary.
>
> Eq. 1 takes a standard tensor decomposition format. It can be motivated by a variety of starting points, e.g. Taylor series approximations. Its hyperparameter is $r$, for which there are standard ways of choosing it, see e.g. [1]-[4].
>
> We address it from two perspectives. *First*, when time is unbounded: there we assume that the true model can be constructed from the exact form in line 149, as motivated by results such as Proposition 1 of [25], which can then be transformed into the structure of Eq. 1. The reasoning is formalized as Proposition 1 in our paper.  *Second*, when time is bounded: we can allow the true data generating functions to belong to a broad class of functions, showing that there exists a finite $r$ that controls the approximation error of Eq. 1 to any a priori degree. This is formalized as Proposition 2 in our paper.
>
> **L136 : ...ANOVA...?**
>
> Analysis of variance. They are the workhorse of analysis of experiments and widely taught across applied sciences.
>
> **L123: … last sentence in this paragraph…**
>
> $k_d$ is the number of parameters associated with a particular level $d$. The symbol was already used in line 114 to describe the time-bounded model. It does not appear explicitly in the time-unbounded model. Since it has three parameters, we define $k_d = 3$, so that we can refer to it later in e.g. Assumption 3 without having to mention which of the intervention models is used.
>
> **Equation in L135**
>
> It is a standard way of writing a regression function with categorical inputs. For instance, if we have two categorical inputs $d_1 \in \{0, 1, 2}; d_2 \in {0, 1}$, we can have the following (overcomplete) ANOVA model with parameters $\theta_{00}, \theta_{01}, \theta_{10}, \theta_{11}, \theta_{20}, \theta_{21}$, with indicator functions $f_{(d_1’, d_2’)}(d_1, d_2) \equiv I(d_1 = d_1’, d_2 = d_2’)$:
>
> $$f(d_1, d_2) = f_{(0, 0)}(d_1, d_2) \times \theta_{00} + \dots + f_{(2, 1)}(d_1, d_2) \times \theta_{21}$$
>
> When there are other inputs beyond the categorical variables of interest, then each $\theta$ can be a function of those.
>
> This appears in many places, e.g. energy functions, exponential families etc., but also all sorts of machine learning methods - as we don’t want to span all exponentially many combinations of the discrete space. In a regression tree with categorical inputs, for instance, we don’t use all combinations, but only those that correspond to paths from the root to a leaf in some tree, with $\theta$ being the piecewise constant expected value of the output at inputs mapped to that leaf. This is a standard interpretation that appears in e.g. Hastie et al.’s *Elements of Statistical Learning*. One interpretation of what we do in Eq.1 is to use a differentiable embedding of the $d$ vector with dimension $r$, instead of $r$ indicator functions given by a tree of $r$ leaves.
>
> **Prop. 3: …equation 1…?**
>
> It does assume that Eq. 1 holds (what would $\beta$ mean otherwise?) The entire Section 2.2. refers to symbols introduced in Eq. 1.
>
> We therefore argue that the clarifications above fully address weakness #1, “The mathematical formalisation...”
>
> **Fig. 1: The assumed causal structure severely limits the applicability … RCT on all units.**
>
> Thanks for raising these points, which quite possibly could come from a few other interested readers too, so it’s very useful to put them to rest.
>
> The structure in Fig. 1 is far more flexible than the standard structure for sequential interventions: it allows interventions to have direct effects indefinitely into the future.
>
> The structure in Fig. 1 does allow confounding: it is only depicting what happens when each $D_t$ is controlled. This is the standard graphical way of denoting an intervention in a random variable $D_t$ (e.g, [22, 32]): wipe the edges into it, adopt a different symbol (here, a square) to indicate that’s a fixed index, not a random variable anymore. This structure is compatible with a non-manipulated graph where the entire past causes $D_t$. It is ill-posed to say that “interventions are assumed to be independent”: intervention variables are not random variables, the concept of probabilistic independence does not apply to them.
>
> We do allow for observational data: the method is introduced *as if* we are allowed to *interpret* each $D_t$ as controlled. To quote line 39,
>
> *“We consider the case where each $D_n^t$ behaves **as if** it was randomized…”*
>
> For avoidance of doubt, in lines 81 and 321:
>
> *"sequentially unconfounded with the system by randomization* **or assumption**"
>
> *“We can rely on standard approaches of sequential ignorability [35] to **justify our method in the absence of randomization**.”*
>
> See also lines 526-532 (Appendix A).
>
> That is, *nothing in our results change at all* if confounding can be blocked by observables, e.g., if in the non-intervened graph we have background variables $Z$ pointing to all $D$, and past $D$ variables pointing to all future $D$ variables (we do not explicitly consider the case of variables $Z^t$ in between variables $D^t$, $D^{t + 1}$, but they can be absorbed into $\phi$ and the do-calculus is still there for us to use.) That is, we can still identify functionals such as Eq. 1, and the conformal prediction results in Section 3.2 do not assume $D$ is randomized. We can make these points more explicitly in the paper, this is a good idea.
>
> We therefore argue that the clarification above fully addresses weakness #2, “The assumptions….”
>
> Thanks again for the questions!

---

> > ### Comment · Reviewer_yF1i · 2024-08-09
> >
> > Thank you for giving a detailed rebuttal. I have read the other reviews, rebuttals and had another look at the paper and will try to answer to this rebuttal below.
> >
> > ## Eq. 1
> >
> > So are you saying that the RHS of Eq. 1 is an estimator for the LHS?
> > The equal sign would suggest that this is a statement about the true data generating process rather than a way to estimate the expectation.
> >
> >
> > ## Fig. 1
> >
> > I'm afraid the explanation did not resolve the confusion.
> >
> > In L39 you say “We consider the case where each $D_n^t$ behaves as if it was randomized…”, a similar quote in L81, which you provide.
> > But in your rebuttal you say that confounding between action variables is allowed, but not preceding action variables causing future actions (and I suppose no confounding between actions and outcomes).
> >
> > Why did you make the assumption of randomization in the first place? In other words, which results in the paper do not hold when you do not make the assumption of randomization?

---

> > > ### Author Response · Authors · 2024-08-09
> > > **Re: Official comment**
> > >
> > > Thank you very much for the reply. We hope the following addresses the remaining follow-up questions.
> > >
> > > **Eq. 1: So are you saying that the RHS of Eq. 1 is an estimator for the LHS? ...**
> > >
> > > Apologies, but we are genuinely confused where this conclusion is coming from. We don't see anything in our rebuttal that suggests that, and it would be helpful to have the quote of the passage leading to it. Eq. 1 is indeed a "a statement about the true data generating process".
> > >
> > > **Fig. 1: ...in your rebuttal you say that confounding between action variables is allowed, but not preceding action variables causing future actions...**
> > >
> > > Again, we don't see anywhere in our rebuttal a statement that $D$ variables cannot be causing others. In fact, we explicitly say the opposite ("*...nothing in our results change at all if ... past $D$ variables pointing to all future $D$ variables.*"). Perhaps it's useful to recall the difference between interventions and random variables within our context.
> > >
> > > In a fully connected DAG according to ordering $(D_1, X_1, D_2, X_2)$ where $D_1$ and $D_2$ are random instead of controlled, we have the edges
> > >
> > > $D_1 \rightarrow X_1, D_1 \rightarrow X_2, D_1 \rightarrow D_2$, $X_1 \rightarrow D_2$, $X_1 \rightarrow X_2$, $D_2 \rightarrow X_2$.
> > >
> > > When $D_1$ is controlled to $d_1$ and $D_2$ to $d_2$, one graphical characterization is
> > >
> > > $d_1 \rightarrow X_1, d_1 \rightarrow X_2$,  $X_1 \rightarrow X_2$, $d_2 \rightarrow X_2$,
> > >
> > > where lower case here indicates that we are talking about exogenous fixed indices (squares in Fig. 1).
> > >
> > > There isn't anything else to be said if $d_2$ is functionally independent of the past, but even then this is not at all an issue. This is because whether $d_2$ is a function or not of $(d_1, X_1)$, it won't affect identification strategies such as the sequential back-door/g-formula see e.g. chapter 4 of [32]. We can still carry $d_2$ symbolically into any averaging over the past, even if $d_2$ is functionally related to $(D_1, X_1)$ and we treat $D_1$ as uncontrolled and average over it (since we don't average over the past, this point is redundant anyway). Other representations such as SWIGs suggest adding edges between fixed indices with functional dependencies see e.g. Chapter 19 of [22], but if we were to adopt SWIGs in Fig. 1 the diagram would become an incomprehensible mess. Technically speaking, even edges like $d_1 \rightarrow X_1$ are "unnecessary", as the graphical model is meant to represent the independence structure of a distribution over random variables, and $d_1$ isn't one (preserving "$d_1 \rightarrow \dots$") saves us from having to label the nodes as $X_1(d_1)$ etc.).
> > >
> > > To summarize, our Fig. 1 is a cosmetic choice among other plausible choices and exemplifies already the case of non-dynamic regimes with no ambiguity. There wasn't really much of a deep point we were trying to make (other than not requiring any Markovian assumptions connecting past and future), and we are earnestly surprised that this is raising a discussion...
> > >
> > > **Why did you make the assumption of randomization in the first place?**
> > >
> > > This was just a way of saying that we assume to have access to the distribution of (single-world) potential outcomes, where our predictions lie. Whether we obtain it by (say) controlled experiments, sequential ignorability, proxies, instrumental variables etc. is orthogonal to our main results. We thought readers would appreciate if we focused on the main novel aspects of our contribution. We are happy to make this point more explicitly in the introduction.
> > >
> > > We really appreciate your engagement, and we hope the above has been helpful.

---

> > > > ### Comment · Reviewer_yF1i · 2024-08-14
> > > >
> > > > Thank you for providing additional details and clarifications.
> > > >
> > > > However, I'm afraid that my main concerns aren't addressed:
> > > >
> > > > - After reading the re-reading the relevant sections in the paper and the rebuttal, it is still unclear to me why Eq. 1 is a reasonable approach to model generic sequences. This also mirrors the overall lack of clarity in the submission.
> > > > - Now, even if we suppose that Eq. 1 was a good ansatz, I think that the assumption of controlling all treatments is overly restrictive for the approach to have an impact for real problems.
> > > >
> > > > Therefore, I will keep my score.

---

> ### Author Response · Authors · 2024-08-14
> **Thank you: closing comment**
>
> Thank you for your assessment. As the discussion started from our paper, we will close it, showing our disagreement,
>
> * "Eq. 1 is a reasonable approach" Eq. 1 follows a matrix-factorisation approach that is well-received by the community. We are still not sure where the misunderstanding is coming from, or how the reviewer initially missed the discussion in Section 2.1, or the source behind statements such as "Eq. 1 as an estimator" following our rebuttal. Perhaps in the same way the reviewer is unfamiliar with ANOVA, they are also unfamiliar with the long history behind structures such as Eq. 1.
>
> *  Predictions under control of a full sequence of actions is a fundamental problem of control theory, dynamic treatment regimes, reinforcement learning, among other fields, including the multi-million user company which gave us feedback on the writing of this paper. It's a literature which we assume our readers feel comfortable with. Once again, we repeat that we do not require the data itself to follow actual randomized trials.
>
> Unfortunately, besides the itemized questions in the review which were responded to in detail above and cross-referenced with the literature and passages in the paper, we do not believe we got any piece of concrete advice that would allow us to address the subjective judgement call of "Wild statements are made without properly introducing the objects, motivating the statement or clarifying what assumptions are made in order for the statement to hold". As such, we would like to leave our own judgement that this statement is unsubstantiated.
>
> To end on a positive note, we are mindful that reviewing is a volunteer, time-consuming job, we are still genuinely thankful for the time and engagement of the reviewer.

---

### Official Review · Reviewer_Ni2f · 2024-07-12

**Soundness:** 3
**Presentation:** 3
**Contribution:** 2
**Rating:** 5
**Confidence:** 3

**Summary:**

The authors propose CSI-VAE, a method that solves the problem of forecasting potential outcomes in multiple interventions. In particular, the authors seem to propose a solution to the problem of a large (combinatorial) space of future treatment plans using the controlled past treatment sequences for inference.

**Strengths:**

STRENGTHS

* The paper is well structured, written, and generally easy to follow.

* Including an automated (distribution free) uncertainty quantification using CP is a nice to have and will definitely improve adoption into practice. in particular in medical applications

* Focusing on identification in this tricky area is very welcome and I would encourage other authors to also include at least a discussion on identification, thank you for this

**Weaknesses:**

WEAKNESSES

* The problem sounds fairly general which would allow other treatment effects over time models to also be relevant to discuss and in particular benchmark against. Currently, it seems the authors only benchmark against ablations of CSI-VAE and naive methods using GRU? Am I mistaken?

**Questions:**

I find the paper to be quite well done, my only remark is wrt the benchmark settings which I would like to see addressed in a rebuttal.

**Limitations:**

see above

---

> ### Author Rebuttal · Authors · 2024-07-31
>
> Thank you for your review, and for finding that our paper is **“quite well done”** and **“generally easy to follow”**.
>
> The question about the benchmark is a good opportunity for further clarifications. In order to keep the paper focused, we stripped away complementary discussions about dealing with confounding to directly address the value of taking a structured approach to causal extrapolation. In the benchmark, this means asking directly whether a black-box method can learn to smooth the gaps in the data by itself, without the structure we build. Several methods could be used. We focused on the GRU because it is a well established method for learning sequential predictions out of categorical input sequences - more flexible than RNNs, less complex than LSTMs or transformers, which would be major overkills: we already overfit with GRUs, a LSTM or beyond would not help (and did not, in preliminary experiments). Autoregressive linear models were also attempted in simpler initial versions of the dataset, but they were already underfitting badly and we did not report them.
>
> We added further methods (LSTMs, transformers) as requested to the shared rebuttal box. Code has been updated (not uploaded yet, we don't think we are allowed to), and it basically replaces calls to GRU with calls to other methods within the PyTorch library. As we anticipated and had seen in preliminary runs, these more complex methods didn't really add much of substance to our central thesis.
>
> Thanks again for your feedback, and happy to take further questions!

---

> > ### Author Response · Authors · 2024-08-13
> > **Closing comments**
> >
> > Thanks again for your review and we hope we have addressed the question about benchmarks - including the easiness by which further comparisons can be added to our codebase.

---

### Author Rebuttal · Authors · 2024-08-05

We take this opportunity to once again thank all reviewers for their time and suggestions! Individualized answers have been provided to each of you.

We use this space to report an update on experiments. To summarize the context, we chose the GRU family merely as an illustration of a modern black-box method that does sequential prediction with representation learning. We didn't think the message would be materially any different if we used a classical RNN, or LSTM, or transformers: the GRU itself was designed to be a good compromise between RNNs and more complex architectures.

In any case, it may be simpler to just run the experiments than argue the above (although we do stress that the main point is structured vs black-box, we never cared whether it was GRU or something else). Since it is straightforward to replace GRUs with other related methods in PyTorch, we did just this. Below are updates including LSTMs and transformers. Autoregressive linear models were also attempted in simpler initial versions of the dataset, but they were already underfitting badly and we do not report them.

*  Fully Synthetic

| Model        | T+1      | T+2      | T+3      | T+4      | T+5      |
| :----------- | :------: | :------: | :------: | :------: | -------: |
| CSI-VAE-1    |  $36.53$ |  $41.46$ |  $41.73$ |  $41.12$ |  $41.32$ |
| CSI-VAE-2    |  $97.80$ | $118.25$ | $117.79$ | $127.25$ | $135.03$ |
| CSI-VAE-3    | $138.78$ | $164.02$ | $141.71$ | $132.59$ | $125.55$ |
| GRU-0        | $229.72$ | $269.66$ | $220.95$ | $208.30$ | $188.43$ |
| GRU-1        | $230.76$ | $270.83$ | $220.93$ | $208.33$ | $184.92$ |
| GRU-2        |  $93.73$ | $101.03$ | $118.01$ |  $88.53$ | $132.28$ |
| LSTM         | $114.71$ | $126.65$ | $137.12$ | $105.22$ | $137.19$ |
| Transformer  | $111.66$ | $122.08$ | $150.57$ | $175.84$ | $87.89$  |

* Semi-Synthetic Spotify

| Model        | T+1      | T+2      | T+3      | T+4      | T+5      |
| :----------- | :------: | :------: | :------: | :------: | -------: |
| CSI-VAE-1    |  $68.23$ |  $82.94$ |  $83.53$ |  $81.97$ | $79.63$  |
| CSI-VAE-2    | $253.85$ | $312.53$ | $305.08$ | $303.68$ | $302.83$ |
| CSI-VAE-3    | $757.94$ | $937.07$ | $800.55$ | $704.66$ | $634.72$ |
|  GRU-0       | $215.42$ | $260.65$ | $193.41$ | $137.20$ | $117.06$ |
| GRU-1        | $223.61$ | $269.69$ | $205.91$ | $141.53$ | $126.36$ |
| GRU-2        | $154.18$ | $187.42$ | $177.96$ | $133.36$ | $127.58$ |
| LSTM         | $130.35$ | $156.02$ | $133.28$ |  $94.35$ |  $85.92$ |
| Transformer  | $133.42$ | $157.66$ | $154.61$ | $164.70$ | $158.03$ |

The attached pdf with box plots provides a visualisation of the above.

---

## Code modifications

Our code will also be updated accordingly (the changes are very localized, basically replacing calls for GRU with other methods already are available in PyTorch).  Changing from GRU to LSTM boils down to commenting a line and adding another:

```
#self.rnn_z_x = nn.GRU(self.z_dim+1, hidden_dim, self.num_layers, batch_first=True)
self.rnn_z_x_d = nn.LSTM(self.z_dim+1+hidden_dim, hidden_dim, self.num_layers, batch_first=True)
```

For transformers, we need to first define a transformer encoder with masking.

```
import torch
import math

class PositionalEncoding(nn.Module):
    def __init__(self, hidden_dim, max_len=5000):
        super().__init__()
        self.hidden_dim = hidden_dim

        pe = torch.zeros(max_len, hidden_dim)
        position = torch.arange(0, max_len, dtype=torch.float).unsqueeze(1)
        div_term = torch.exp(torch.arange(0, hidden_dim, 2).float() * (-math.log(10000.0) / hidden_dim))
        pe[:, 0::2] = torch.sin(position * div_term)
        pe[:, 1::2] = torch.cos(position * div_term)
        pe = pe.unsqueeze(0).transpose(0, 1)
        self.register_buffer('pe', pe)

    def forward(self, x):
        # x of size B, T, H
        B, T, _ = x.size()
        # print(x.size())
        # print(self.pe.size())
        pe = torch.permute(self.pe, (1, 0, 2))
        # print(self.pe.size())
        x = x + pe[:, :T, :].expand(B, -1, -1)
        # print(x.size())
        # x += self.pe
        return x

class TransformerTimeSeries(nn.Module):
    def __init__(self, hidden_dim, nhead, num_layers=1):
        super().__init__()
        self.hidden_dim = hidden_dim
        self.pos_encoder = PositionalEncoding(hidden_dim)

        encoder_layers = nn.TransformerEncoderLayer(hidden_dim, nhead, dim_feedforward=hidden_dim*4, batch_first=True)
        self.transformer_encoder = nn.TransformerEncoder(encoder_layers, num_layers)

    def generate_square_subsequent_mask(self, sz):
        mask = (torch.triu(torch.ones(sz, sz)) == 1).transpose(0, 1)
        mask = mask.float().masked_fill(mask == 0, float('-inf')).masked_fill(mask == 1, float(0.0))
        return mask

    def forward(self, x):
        '''
        input:
        x of size (B, T, hidden)
        output:
        x_T of size (B, T, hidden)
        '''
        x = self.pos_encoder(x)
        _, T, _ = x.size()
        tgt_mask = self.generate_square_subsequent_mask(T).to(x.device)
        # print(tgt_mask)

        h = self.transformer_encoder(x, mask=tgt_mask, is_causal=True)

        return h
```
 and then modify the call as in

```
# self.rnn_z_x = nn.GRU(self.z_dim+1, hidden_dim, self.num_layers, batch_first=True)
self.rnn_z_x_d = TransformerTimeSeries(self.z_dim+1+hidden_dim, nhead=8)
```

Many thanks again!

---

### Decision · Program_Chairs · 2024-09-25

**Decision:**

Accept (poster)

**Comment:**

The submission addresses learning a structural model for sequential treatment regimes where interventions are composed over time. Theoretical results, a VAE based algorithm and experimental validation are provided.

While most reviewers consider the contribution to be sound and valuable for the field, there has been concerns that the manuscript is hard to parse. Overall, the contribution is sound and significant, but highly technical, such that it is hard to follow beyond a community with strong expertise in statistical models for potential outcomes. Given the significance and strengths of the contribution, the AC suggests acceptance of the paper as a poster. However, the AC recommends to make an effort towards the general readership by giving more insights into their model in the camera ready version, e.g. by exposing early in the manuscript an example in a simplified setting.